# Learning Multi-Scale Hypergraph for High-Order Brain Connectivity Analysis

Jaeyoon Sim [1]    Soojin Hwang [1]    Seunghun Baek [1]    Guorong Wu [2]    Won Hwa Kim [1]

## Abstract

Understanding complex interactions between brain regions is critical for early neurodegenerative disease classification such as Alzheimer's Disease (AD) and Parkinson's Disease (PD). While graph-based models are widely used to analyze brain networks, most existing approaches primarily focus on pairwise interactions between directly connected nodes, limiting their ability to capture higher-order dependencies across multiple regions. Although hypergraph-based methods have been proposed to model higher-order relations, many rely on predefined hyperedges or restrict learning to hyperedge weights, reducing flexibility and limiting their capacity to capture multi-resolution structural patterns. In this regard, we introduce an adaptive multi-scale hyperedge learning framework, i.e., MuHL, which constructs hierarchical node features and dynamically learns high-order interactions through continuous hyperedge construction over multi-resolution graph signals. Extensive experiments on multiple brain network benchmarks demonstrate that MuHL consistently improves disease classification performance across different stages, and further identifies key regions of interest (ROIs) and their groupwise interactions from the learned hyperedges that are associated with disease progression, highlighting its potential as a powerful tool for brain network analysis in neurodegenerative disorders.

## 1. Introduction

Capturing relationships of multiple variables beyond one-to-one interaction is an important problem in various domains such as social network analysis (e.g., modeling social groups) (Benson et al., 2016; Zhu et al., 2018; Battiston et al., 2020), computer vision (e.g., scene graph generation) (Zellers et al., 2018; Johnson et al., 2018) and systems biology (e.g., complexes of proteins) (Pavlopoulos et al., 2011; Hu et al., 2021; Feng et al., 2021). Similarly, in neuroimaging, characterizing high-order interactions between brain regions of interest (ROIs) is crucial for early neurodegenerative disease detection (Han et al., 2025). Typically, a complex wiring representation of the brain as a brain network, e.g., structural connectivity from Diffusion Tensor Image (DTI) or functional connectivity from functional MRI, is derived to investigate the interactions between ROIs.

For example, degenerative disorders such as Alzheimer's disease (AD) and Parkinson's disease (PD) are characterized by disruptions in structural brain connectivity (Matthews et al., 2012; Reid & Evans, 2013) as well as regional abnormalities, including cortical atrophy and reduced brain volume (Thompson et al., 2007; Mrdjen et al., 2019). These changes reflect spatially distributed neuronal degeneration and have been widely adopted as imaging biomarkers for early diagnosis and disease progression monitoring. Modern neuroimaging modalities provide complementary views of these disease-related alterations. For instance, Diffusion Weighted Imaging (DWI) reveals connectivity disruptions by enabling the construction of structural brain networks from white matter fiber pathways, while region-wise measures derived from Magnetic Resonance Imaging (MRI) and Positron Emission Tomography (PET) offer a comprehensive multimodal characterization of disease-specific structural and functional changes (Baek et al., 2024a;b).

Given the complex structure of brain connectivity, graph-based models provide an effective way to analyze these multi-sensing features by representing both connectivity structures and regional characteristics (Cui et al., 2022a;b; Kan et al., 2022; Zhou et al., 2024; Sim et al., 2024a;b; Cho et al., 2024). However, existing methods based on conventional graph convolution typically neglect higher-order relationships among brain regions (Kipf & Welling, 2016; Veličković et al., 2018; Chen et al., 2020b), and stacking graph convolution layers indirectly consider these interactions at the cost of oversmoothing problem (Chen et al., 2020a). Therefore, standard graph models, relying on pairwise interactions denoted by adjacency, struggle to capture group-wise dependencies effectively and have difficulties in modeling the complex topological changes associated with

[1]Pohang University of Science and Technology (POSTECH), Pohang, South Korea [2]University of North Carolina at Chapel Hill, Chapel Hill, USA. Correspondence to: Won Hwa Kim <wonhwa@postech.ac.kr>.

*Proceedings of the 43rd International Conference on Machine Learning*, Seoul, South Korea. PMLR 306, 2026. Copyright 2026 by the author(s).

neurodegenerative disorders. Such limitations are especially critical for brain networks, where functional and structural interactions often arise from the coordinated activity of multiple regions. Capturing these group-wise dependencies is essential for characterizing structural and functional patterns that reflect disease- related alterations in brain organization.

In this regime, a more flexible framework is required to analyze complex brain networks and reveal higher-order relationships among regions of interest (ROIs) associated with specific disease patterns. We approach this problem from two complementary perspectives. First, we learn adaptive *multi-resolution* graph representations from a graph wavelet perspective that encode multi-order node-wise neighborhood information. Second, we construct a *hypergraph* in which hyperedges connect multiple nodes simultaneously, enabling explicit modeling of high-order group interactions that are not represented in the original graph.

Our multi-resolution representation captures ROI characteristics across multi-scale spatial resolutions through diffusion scales in spectral graph filtering. Different diffusion levels control the extent of node smoothing, forming coarse-to-fine neighborhoods within a single atlas and naturally expanding the effective localization range of each ROI without requiring multiple parcellations. Building upon these representations, the hypergraph allows us to capture disease-specific patterns arising from interactions among multiple ROIs. Unlike prior hypergraph methods that rely on predefined hyperedges or only adjust their weights, we learn continuous, scale-aware hyperedges directly from multi-resolution features. Based on these ideas, we propose **Mu**lti-scale **H**yperedge **L**earning (MuHL).

**The key contributions of our work** are:

**i)** Proposing a novel framework which dynamically learns rich and implicit group-wise node dependencies (i.e., hyperedges),

**ii)** Adaptively capturing structural variations at different scales for hypergraph construction to enhance downstream classification,

**iii)** Demonstrating theoretical insights that the adaptive hyperedges are derived by modulating node features in the graph wavelet space.

Extensive validation was performed on two independent public benchmarks: the Alzheimer's Disease Neuroimaging Initiative (ADNI), using structural networks from DTI and ROI measurements from functional imaging, and the Parkinson's Progression Markers Initiative (PPMI), using fMRI-derived Blood-Oxygen-Level-Dependent (BOLD) signals, which demonstrates the superiority of MuHL over state-of-the-art methods while providing biological interpretability of brain networks by identifying key sub-structure based on the learned hyperedges.

## 2. Related Works

**High-order Graph Learning.** The vanilla GCN (Kipf & Welling, 2016) and variants of Graph Neural Networks (GNNs) utilize graph convolution to perform feature aggregation from neighbors. GCNII (Chen et al., 2020b) extends a vanilla GCN with residual connection and identity mapping, and Graph Attention Network (GAT) (Veličković et al., 2018) introduces an effective attention mechanism on graphs to assign relationships to neighboring nodes. These works primarily rely on pairwise interactions and local message passing, which makes it difficult to capture rich group-wise dependencies in complex graphs (e.g., structural/functional brain connectomes).

Several studies have incorporated higher-order structures into graphs to better capture complex geometric relationships. In particular, hypergraphs have been widely adopted to model group-wise interactions among multiple nodes. For example, HGNN (Feng et al., 2019) captures high-order correlations using hypergraph representations, HNHN (Dong et al., 2020) introduces nonlinear activations on both nodes and hyperedges, UniGCNII (Huang & Yang, 2021) mitigates over-smoothing in hypergraph learning via residual connections and identity mapping, HyperDrop (Jo et al., 2021) employs hypergraph-based edge pooling to remove task-irrelevant edges, and HyperGT (Liu et al., 2024) encodes both global and local structure by integrating a transformer architecture with a hypergraph. Recently, DHHNN (Mei et al., 2025) proposes a dynamic hyperbolic hypergraph neural network that iteratively updates hyperedge weights through hyperbolic attention during training.

**Brain Network Analysis.** Recent works have been designed to analyze the brain networks. They adopted or developed GNN and its variants, as the nodes (i.e., ROIs) within the same brain template have distinct locations and unique identities. IBGNN (Cui et al., 2022b) proposes an interpretable framework to analyze disorder-specific ROIs and prominent connections, and BrainGB (Cui et al., 2022a) standardizes the process by summarizing brain network construction pipelines and modularizing GNN designs, and SGCN (Zhou et al., 2024) proposes an interpretable framework by defining the importance probability of multiple brain imaging data. Notably, BrainNetTF (Kan et al., 2022) learns fully pairwise attentions with transformer-based models, ALTER (Yu et al., 2024) also adopts a transformer with a biased random walk to capture long-range dependencies, BioBGT (Peng et al., 2025) captures the small-world architecture of brain graphs using network entanglement, and Brain Quadratic Network (BQN) (Yang et al., 2025) adopts quadratic networks for brain network analysis.

Beyond conventional graph-based approaches, recent studies have explored hypergraph-based frameworks to better capture higher-order interactions in brain networks.

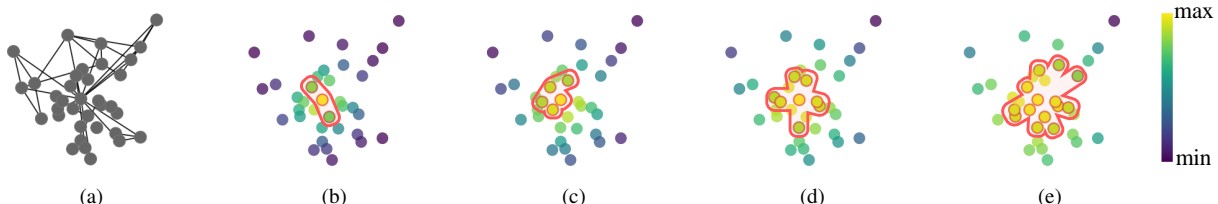

*Figure 1.* Visualization of multi-resolution high-order relation modeling via hypergraph on a random graph. The original graph with pairwise edges is shown in (a), and from fine scale (b) to coarse scale (e), hyperedges adaptively expand to include more nodes, moving from localized connections to broader groupings that capture global structural similarity. Similar colors indicate potentially similar regions, while the red contour represents the sparsely constructed hyperedge at each scale, determined by the corresponding smoothed features.

Dynamic Weighted Hypergraph Convolutional Network (dwHGCN) (Wang et al., 2023) focuses on learning weights for predefined hyperedges to characterize high-order relations among brain regions, and HyBRiD (Qiu et al., 2024) constructs discrete hyperedges by sampling binary masks and assigning a learnable weight to each hyperedge. While previous works primarily optimize hyperedge weights, our method directly refines connectivity patterns through adaptive soft hyperedges derived from multi-scale spectral diffusion, providing a more flexible and hierarchically expressive representation of connectivity.

## 3. Preliminaries

**Hypergraph.** An undirected graph $\mathcal{G}=\{\mathcal{V}, \mathcal{E}\}$ with $N$ nodes comprises a node set $\mathcal{V}$ and an edge set $\mathcal{E}$. A symmetric adjacency matrix $\mathcal{A} \in \mathbb{R}^{N \times N}$ and a diagonal matrix $\mathcal{D} \in \mathbb{R}^{N \times N}$ are constructed from $\mathcal{E}$, encoding connectivity among nodes and the volume of each node, respectively. Here, a graph Laplacian $\mathcal{L}=\mathcal{D}-\mathcal{A}$ is real and positive semi-definite. It has a complete set of orthonormal basis $U=[u_1|u_2|\ldots|u_N]$ known as Laplacian eigenvectors and corresponding real and non-negative eigenvalues $0=\lambda_1 \leq \lambda_2 \ldots \leq \lambda_N$, so does the normalized Laplacian $\hat{\mathcal{L}}=\mathcal{D}^{-1/2}\mathcal{L}\mathcal{D}^{-1/2}$.

A hypergraph with $N$ nodes and $M$ hyperedges consists of $\mathcal{V}$ and a hyperedge set $\mathcal{E}'$. It is a generalization of a conventional graph where each hyperedge represents group-wise interactions by connecting a set of nodes simultaneously. The structure of hypergraph can be represented as an incidence matrix $H \in \mathbb{R}^{N \times M}$, where each entry $H(v_n, e_m)$ is defined as 1 if node $v_n$ belongs to edge $e_m$, and 0 otherwise, for all $v_n \in \mathcal{V}$ and $e_m \in \mathcal{E}'$. The degree of $n$-th node is defined as $d(v_n)=\sum_{m=1}^{M} H(v_n, e_m)$ and the degree of the $m$-th hyperedge is defined as $d(e_m)=\sum_{n=1}^{N} H(v_n, e_m)$. Accordingly, the node and hyperedge degrees are represented in diagonal matrices $\mathcal{D}_v \in \mathbb{R}^{N \times N}$ and $\mathcal{D}_e \in \mathbb{R}^{M \times M}$, respectively.

**Multi-scale via Graph Wavelet Transform.** Spectral Graph Wavelet Transform (SGWT) (Hammond et al., 2011) extends traditional wavelet transform (Mallat, 1999) to graphs via spectral graph theory (Chung, 1997). Using SGWT, a graph signal $x$ is decomposed into multiple granularities $x_s$ to facilitate multi-resolution analysis. To project the signals into the spectral domain, SGWT uses wavelet basis $\psi_s = Ug(s\Lambda)U^T$, where $g(\cdot)$ denotes a kernel with the scale $s$. The $s$ controls the bandwidth of locality in the graph Fourier space, capturing either global structure or local details at specific resolutions. Projecting the signal $x$ into the spectral domain via these bases yields the wavelet coefficient $W_x(s) = \psi_s \cdot x$. Under the admissibility condition (Mallat, 1999), the inverse transform perfectly reconstructs original $x$ as $\frac{1}{C_k}\int_0^\infty \psi_s \cdot W_x(s)\frac{ds}{s}$, with an admissibility constant $C_k = \int_0^\infty \frac{g(\lambda)^2}{\lambda}d\lambda < \infty$. It represents a superposition of multi-resolution representations of $x$ over scales $s \in [0, \infty)$. Thus, a signal $x_s$ in the graph space filtered at the specific scale $s$ is defined as

$$x_s = \psi_s \cdot W_x(s) = Ug^2(s\Lambda)U^T x, \tag{1}$$

which allows extracting signals at a specific resolution. In our work, we make these scales learnable to obtain local-to-global characteristics adaptively.

## 4. Adaptive Multi-Scale Hypergraph Learning

**Overview of High-order Structure Learning.** We introduce MuHL, a unified framework that jointly performs multi-resolution graph wavelet filtering and hypergraph learning to capture high-order structural dependencies between the multi-scale features. Instead of treating multi-resolution filtering and hyperedge modeling as disjoint processes, MuHL tightly couples them to support learning expressive high-order structures by dynamically constructing scale-specific hyperedges.

Fig. 1 illustrates how multi-resolution wavelet filtering influences the formation of scale-specific hyperedges by modulating node similarity through varying receptive fields. Compared to the pair-wise graph in (a), the wavelet kernel in small scales operates over a narrow receptive field, resulting in localized and distinctive features, as in Fig. 1 (b). This leads to the formation of compact hyperedges that connect only a few strongly correlated nodes. In contrast, Fig. 1 (e) depicts a larger receptive field induced by a higher-scale kernel, which smooths features across a broader region and increases similarity among distant nodes. Consequently, more nodes are grouped together into larger hyperedges, reflecting expanded high-order relationships. This progressive

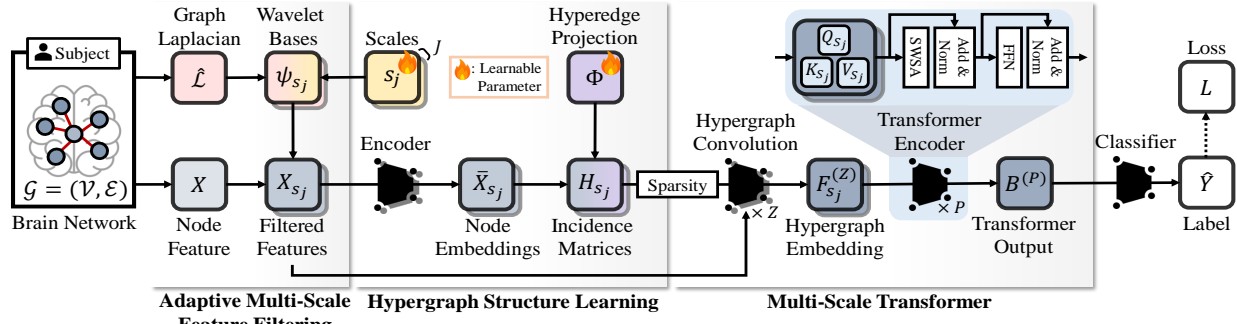

*Figure 2.* Illustration of MuHL. A brain network $\mathcal{G}$ with node features $X$ goes through 1) Adaptive multi-scale feature filtering to produce $X_s$, 2) Hypergraph structure learning to estimate incidence matrix of a hypergraph $H_s$, and 3) Multi-scale transformer to yield $B^{(P)}$, incorporating all scale-wise information from the hyperedges. Finally, a classifier predicts a disease-specific label for $\mathcal{G}$. Both the $s$ and $\Phi$ are optimized during the training.

expansion across multiple scales enables MuHL to learn a hierarchy of high-order hyperedges, each reflecting different affinities. By modeling interactions at both fine and coarse structural levels, MuHL builds an expressive hypergraph that jointly encodes local variation and global consistency.

**Adaptive Multi-Scale Feature Filtering.** Consider a graph $\mathcal{G}$ as a normalized Laplacian $\hat{\mathcal{L}}$ and a set of features $X \in \mathbb{R}^{N \times D}$ (e.g., $D$ imaging measures on $N$ ROIs), and a label $Y$. Given trainable scales $\{s_j\}_{j=0}^{J}$, MuHL extracts task-relevant graph information at specific resolutions. Fig. 2 shows that $X$ is decomposed into multi-resolution components $X_{s_j}$ across various scales using Eq. (1), which is iteratively updated within MuHL. In this step, the input features $X$ are passed through one low-pass and several band-pass wavelet kernels, where each kernel extracts structural information at a different resolution. The resulting scale-wise representations $X_{s_j}$ capture global-to-local variations and are passed to the hypergraph learning framework.

**Multi-Scale Hypergraph Structure Learning.** The multi-resolution representations $\{X_{s_j}\}_{j=1}^{J}$ are used to generate incidence matrices to model group-wise node interactions across scales. Given the $X_{s_j}$ at $j$-th scale, a node embedding $\bar{X}_{s_j} \in \mathbb{R}^{N \times d_h}$ is obtained via a linear transformation, where $d_h$ is the hidden feature dimension. To obtain a flexible hypergraph structure, we introduce a learnable hyperedge projection matrix $\Phi \in \mathbb{R}^{d_h \times M}$, where $M$ is the number of hyperedges. Using $\Phi$, the scale-specific incidence matrix is computed as $H_{s_j} = \bar{X}_{s_j}\Phi$, which encodes node-to-hyperedge affinities at scale $j$. This enables each scale to capture its own group-level ROI interactions.

**Proposition 1.** *Given a graph signal $X$ and a hyperedge projector $\Phi$, computing incidence matrix $H_s$ is equivalent to projecting the graph wavelet representation of $X$, i.e., $W_X(s)$, with $\Phi$.*

The proposition 1 tells that the multi-resolution hypergraph structure can be characterized through the graph wavelet representation in the graph frequency space, whose concept

is ambiguous in the original graph space due to the irregular graph structure. This further leads to the following proposition, i.e., dilation property of the learned hypergraph that, as the wavelet scale increases, the smoothing effect on node features enlarges hyperedges by increasing the cardinality of the node set associated with the hyperedge.

**Proposition 2.** *In $H_s$, there exists at least one hyperedge $e_m^*$ such that the number of nodes within $e_m^*$, for which $e_m^*$ attains the highest weight, monotonically increases with respect to $s$, and this number converges to $N$ as $s \to \infty$, i.e., $\lim_{s \to \infty} |\{v_n : e_m^* = \arg\max_{e_m} H_s(v_n, e_m)\}| \approx N$.*

Consequently, the smoothness of node features at each scale directly drives the expansion of hyperedges, providing the theoretical basis for the progressive enlargement of hyperedges observed in Fig. 1, where localized connections at fine scales gradually evolve into broader groupings at coarser scales. The detailed proofs are provided in Appendix A.

To further refine $H_s$, we convert the dense incidence values into a normalized form, yielding a normalized incidence matrix at scale $s_j$ as

$$\tilde{H}_{s_j} = \text{SoftMax}(\text{ReLU}(\bar{X}_{s_j}\Phi)), \qquad (2)$$

where the $\text{ReLU}$ activation eliminates weak connections among hyperedges and the $\text{SoftMax}$ function is applied to normalize each row of $\tilde{H}_{s_j}$ into a probability distribution over hyperedges.

To prevent excessive connections, $\tilde{H}_{s_j}$ is made sparse by limiting the number of neighboring hyperedges per node to $\eta$, resulting in a truncated incidence matrix $\bar{H}_{s_j}$ as

$$\bar{H}_{s_j}(n,m) = \begin{cases} \tilde{H}_{s_j}, & \tilde{H}_{s_j}(n,m) \in \text{TopK}(\tilde{H}_{s_j}(n,*),\eta) \\ 0, & \tilde{H}_{s_j}(n,m) \notin \text{TopK}(\tilde{H}_{s_j}(n,*),\eta) \end{cases}$$
$$(3)$$

where $\text{TopK}$ is a top-$k$ function. Adaptively acquiring these incidence matrices enables capturing higher-order dependencies among nodes across scales and model a wide range of implicit interactions without any predefined rules.

**Multi-Scale Transformer.** Building on the constructed multi-scale hypergraphs, MuHL extracts scale-wise features by performing hypergraph convolution on the paired inputs $\{X_{s_j}\}_{j=1}^J$ and $\{\bar{H}_{s_j}\}_{j=1}^J$ through encoders in the transformer block, capturing structural patterns unique to each scale. Based on Feng et al. (2019), scale-wise hypergraph convolution operation at $z$-th layer is given as

$$F_{s_j}^{(z)} = \sigma(\mathcal{D}_v^{-1/2} \bar{H}_{s_j} W_e \mathcal{D}_e^{-1} \bar{H}_{s_j}^T \mathcal{D}_v^{-1/2} F_{s_j}^{(z-1)} \Theta^{(z)}), \quad (4)$$

where $F_{s_j}^{(z)}$ is the output at $j$-th scale with $F_{s_j}^{(0)} = X_{s_j}$, $W_e$ is a diagonal matrix representing the weights of hyperedges, $\Theta^{(z)}$ is the learnable weight at $z$-th layer, and $\sigma$ denotes an activation function. Since Eq. (4) is an operation in a single convolution layer, we can stack $Z$ of them to achieve better representations across all resolutions.

Then, as in Fig. 2, the $\{F_{s_j}^{(Z)}\}_{j=1}^J$ are fed into an attention layer to compute node-wise attention scores. We introduce the Scale-Wise Self-Attention (SWSA) module, where each head focuses on a specific scale and operates over a graph structure to integrate information from other nodes. The input of SWSA consists of $d_k$-dimensional query $Q_{s_j} \in \mathbb{R}^{N \times d_k}$, key $K_{s_j} \in \mathbb{R}^{N \times d_k}$, and value $V_{s_j} \in \mathbb{R}^{N \times d_k}$ from $F_{s_j}^{(Z)}$. Then, the SWSA scores are computed as

$$A_{s_j} = \text{Softmax}(\frac{Q_{s_j} K_{s_j}^T}{\sqrt{d_k}}) \in \mathbb{R}^{N \times N}, \quad (5)$$

where $\text{Softmax}$ function normalizes the similarity scores to obtain attention weights. These weights indicate the relative influence of each node within the graph at scale $s_j$. Using them, the self-attention value at the $j$-th scale is computed as $\phi(Q_{s_j}, K_{s_j}, V_{s_j}) = A_{s_j} V_{s_j}$. As each head is assigned to a single scale, global features across scales are averaged with a multi-attention function as

$$\text{SWSA}(Q, K, V) = [h^{(1)}|\ldots|h^{(j)}|\ldots|h^{(J)}]W_O,$$
$$h^{(j)} = \phi(Q_{s_j} W_{Q_{s_j}}, K_{s_j} W_{K_{s_j}}, V_{s_j} W_{V_{s_j}}), \quad (6)$$

where $W_{Q_{s_j}}$, $W_{K_{s_j}}$, $W_{V_{s_j}}$, and $W_O$ are projection matrices for $Q_{s_j}$, $V_{s_j}$, $K_{s_j}$, and SWSA respectively. Thus, SWSA enables the model to jointly attend to information from different scales across various ROIs in long-range. In addition to SWSA, each layer from the transformer encoder contains a fully connected feed-forward network (FFN). To stabilize the training of MuHL, residual connections (He et al., 2016) and layer normalization (LN) (Ba et al., 2016) are employed. This configuration effectively captures comprehensive context across all nodes as

$$B^{(p)} = \text{LN}(\tilde{B}^{(p)} + \text{FFN}(\tilde{B}^{(p)})),$$
$$\tilde{B}^{(p)} = \text{LN}(B^{(p-1)} + \text{SWSA}(B^{(p-1)})), \quad (7)$$

where $B^{(p)}$ is an output from $p$-th attention layer, and multi-scale representations $\{F_{s_j}^{(Z)}\}_{j=1}^J$ are used as $Q$, $K$ and $V$

for $B^{(0)}$. Also, we can stack $P$ attention layers to capture complex dependencies in the input features.

**Training Objective.** Given a set of graphs $\{\mathcal{G}_t\}_{t=1}^T$ with corresponding labels $\{Y_t\}_{t=1}^T$, we train a classifier such that $f(B_t^{(P)}) = Y_t$, which takes the $B^{(P)}$ as an input from Eq. (7) and returns a prediction $\hat{Y}_{tc}$ for the $c$-th class of sample $t$. To update all parameters, the objective function is defined by cross-entropy between the true value $Y_{tc}$ and $\hat{Y}_{tc}$. With an $l_1$-norm regularization on a scale $s_j$ to ensure its positivity, the overall objective function $L$ is defined as

$$L = -\frac{1}{T}\sum_{t=1}^T \sum_{c=1}^C Y_{tc}\log(\hat{Y}_{tc}) + \alpha\frac{1}{J}\sum_{j=1}^J \mathbf{1}_{s<0}|s_j|, \quad (8)$$

where $\alpha$ is a user-parameter and $\mathbf{1}$ is an indicator function. The loss $L$ is minimized via backpropagation to optimize learnable parameters, including the scale $s_j$ and the hyperedge weight $E_{s_j}$ at $j$-th level, over the full scale range $J$.

## 5. Experiments

**Dataset.** To evaluate our framework, we utilized two independent neurodegenerative brain network benchmarks: 1) Alzheimer's Disease Neuroimaging Initiative (ADNI) (Mueller et al., 2005) and 2) Parkinson's Progression Markers Initiative (PPMI) (Marek et al., 2011).

*ADNI.* Neuroimages of 650 subjects in the ADNI study were processed using in-house pipeline. Each brain was partitioned into 148 cortical and 12 sub-cortical regions by Destrieux atlas (Destrieux et al., 2010), and Tractography on DWI was applied to derive white matter fibers connecting the 160 brain regions. For each parcellation, region-wise imaging features were defined such as Standard Uptake Value Ratio (SUVR) (Thie, 2004) of metabolism (FDG), $\beta$-amyloid protein (AMY), and tau protein (TAU) from PETs, along with cortical thickness (CT) from MRI scans, were measured. Each subject was assigned to Control (CN, $T=226$), Significant Memory Concern (SMC, $T=131$), Early/Late Mild Cognitive Impairment (EMCI/LMCI, $T=217/64$), and AD ($T=12$) groups.

*PPMI.* Functional MRI data from 181 subjects in the PPMI study were provided by (Marek et al., 2011), and subsequently preprocessed into brain networks by (Xu et al., 2023). Each brain was parcellated into 116 regions using the AAL atlas (Tzourio-Mazoyer et al., 2002), from which region-wise Blood-Oxygen-Level-Dependent (BOLD) signals were extracted as node features. Functional connectivity matrices were constructed by computing pairwise correlations between regional BOLD signals and were used as edge features. Each subject was classified into three groups: Cognitively Normal (CN, $T=15$), Prodromal ($T=53$), and PD ($T=113$), reflecting a progression from CN to PD.

*Table 1.* Brain network classification performance on the ADNI and PPMI datasets evaluated using 5 fold cross-validation. The best results are shown in **bold**, and the second-best results are underlined. (†: Methods for brain network analysis)

| Category | Method | ADNI (CN vs SMC vs EMCI vs LMCI vs AD) | | | | PPMI (CN vs Prodromal vs PD) | | | |
|---|---|---|---|---|---|---|---|---|---|
| | | Acc ↑ | Pre ↑ | Rec ↑ | F1s ↑ | Acc ↑ | Pre ↑ | Rec ↑ | F1s ↑ |
| Graph based method | GCN (Kipf & Welling, 2016) | $85.7 \pm 1.7$ | $88.1 \pm 3.0$ | $82.7 \pm 5.9$ | $83.7 \pm 3.4$ | $66.3 \pm 2.4$ | $46.2 \pm 5.4$ | $41.5 \pm 3.7$ | $38.9 \pm 4.4$ |
| | GAT (Veličković et al., 2018) | $89.2 \pm 2.1$ | $92.7 \pm 1.8$ | $86.6 \pm 4.6$ | $88.9 \pm 3.6$ | $\underline{72.9 \pm 2.5}$ | $55.1 \pm 7.8$ | $52.0 \pm 3.3$ | $52.3 \pm 4.3$ |
| | GCNII (Chen et al., 2020b) | $86.3 \pm 4.0$ | $91.2 \pm 1.5$ | $85.8 \pm 6.6$ | $87.4 \pm 4.8$ | $68.5 \pm 2.3$ | $52.5 \pm 6.0$ | $42.8 \pm 4.5$ | $41.4 \pm 6.3$ |
| | BrainGNN† (Li et al., 2021) | $43.4 \pm 4.7$ | $31.8 \pm 3.4$ | $28.3 \pm 1.7$ | $27.0 \pm 1.3$ | $68.1 \pm 2.0$ | $48.8 \pm 5.3$ | $40.8 \pm 4.4$ | $38.2 \pm 5.3$ |
| | IBGNN† (Cui et al., 2022b) | $81.4 \pm 2.4$ | $85.8 \pm 0.7$ | $82.3 \pm 3.4$ | $83.3 \pm 2.3$ | $68.1 \pm 2.4$ | $51.1 \pm 6.0$ | $42.1 \pm 5.2$ | $38.7 \pm 5.4$ |
| | BrainGB† (Cui et al., 2022a) | $83.5 \pm 1.6$ | $87.1 \pm 2.6$ | $80.7 \pm 3.2$ | $82.2 \pm 2.3$ | $68.8 \pm 3.3$ | $48.8 \pm 6.0$ | $42.8 \pm 3.2$ | $41.6 \pm 4.5$ |
| | BrainNetTF† (Kan et al., 2022) | $89.4 \pm 2.5$ | $88.9 \pm 5.7$ | $84.6 \pm 7.3$ | $85.8 \pm 6.0$ | $67.9 \pm 2.9$ | $55.6 \pm 8.8$ | $45.7 \pm 5.1$ | $45.0 \pm 7.2$ |
| | SGCN† (Zhou et al., 2024) | $84.0 \pm 2.1$ | $87.6 \pm 3.3$ | $82.1 \pm 4.6$ | $83.8 \pm 3.5$ | $67.6 \pm 3.2$ | $39.0 \pm 9.3$ | $40.6 \pm 4.4$ | $37.6 \pm 6.8$ |
| | ALTER† (Yu et al., 2024) | $\underline{90.8 \pm 2.6}$ | $\underline{93.3 \pm 2.4}$ | $\underline{89.6 \pm 5.6}$ | $\underline{90.9 \pm 4.6}$ | $69.6 \pm 2.3$ | $55.2 \pm 8.5$ | $\underline{55.8 \pm 5.3}$ | $54.5 \pm 8.3$ |
| | BioBGT† (Peng et al., 2025) | $80.3 \pm 1.7$ | $79.7 \pm 7.1$ | $77.3 \pm 6.7$ | $77.2 \pm 6.1$ | $68.0 \pm 2.8$ | $54.3 \pm 8.2$ | $48.7 \pm 4.2$ | $45.7 \pm 3.2$ |
| | BQN† (Yang et al., 2025) | $81.5 \pm 4.7$ | $74.3 \pm 5.6$ | $72.3 \pm 5.5$ | $72.6 \pm 5.2$ | $71.9 \pm 2.9$ | $\underline{63.2 \pm 6.2}$ | $51.0 \pm 8.4$ | $\underline{56.4 \pm 7.3}$ |
| Hypergraph based method | HGNN (Feng et al., 2019) | $83.2 \pm 2.7$ | $84.8 \pm 7.1$ | $80.7 \pm 6.8$ | $82.1 \pm 6.6$ | $65.2 \pm 4.8$ | $43.3 \pm 9.0$ | $41.9 \pm 6.1$ | $40.3 \pm 7.1$ |
| | HNHN (Dong et al., 2020) | $87.3 \pm 1.8$ | $86.9 \pm 1.0$ | $85.6 \pm 4.2$ | $87.0 \pm 3.0$ | $69.1 \pm 2.9$ | $52.6 \pm 5.1$ | $48.5 \pm 8.4$ | $48.0 \pm 9.4$ |
| | UniGCNII (Huang & Yang, 2021) | $89.6 \pm 1.6$ | $91.1 \pm 1.8$ | $88.0 \pm 3.8$ | $89.9 \pm 2.0$ | $70.8 \pm 3.3$ | $59.6 \pm 9.4$ | $54.3 \pm 7.2$ | $55.2 \pm 7.3$ |
| | HyperDrop (Jo et al., 2021) | $72.3 \pm 0.8$ | $74.4 \pm 3.1$ | $65.9 \pm 0.2$ | $68.6 \pm 2.2$ | $67.5 \pm 4.0$ | $41.5 \pm 8.9$ | $39.2 \pm 3.8$ | $36.0 \pm 6.4$ |
| | dwHGCN† (Wang et al., 2023) | $90.2 \pm 1.3$ | $87.2 \pm 7.1$ | $86.3 \pm 6.8$ | $86.2 \pm 6.4$ | $69.0 \pm 2.9$ | $51.5 \pm 7.9$ | $44.2 \pm 6.1$ | $43.3 \pm 8.0$ |
| | HyperGT (Liu et al., 2024) | $81.5 \pm 1.6$ | $92.1 \pm 1.8$ | $89.0 \pm 3.8$ | $89.9 \pm 2.0$ | $68.5 \pm 1.5$ | $51.7 \pm 4.4$ | $42.0 \pm 4.8$ | $39.6 \pm 6.1$ |
| | HyBRiD† (Qiu et al., 2024) | $86.6 \pm 4.2$ | $87.5 \pm 3.9$ | $84.3 \pm 6.4$ | $84.6 \pm 4.7$ | $67.2 \pm 6.5$ | $54.8 \pm 6.8$ | $52.0 \pm 6.2$ | $52.8 \pm 6.5$ |
| | DHHNN (Mei et al., 2025) | $81.1 \pm 2.5$ | $89.0 \pm 4.5$ | $84.5 \pm 3.2$ | $86.7 \pm 3.4$ | $60.6 \pm 3.8$ | $62.3 \pm 9.8$ | $52.7 \pm 4.3$ | $54.0 \pm 6.3$ |
| | MuHL (Ours) | $\mathbf{93.2 \pm 2.4}$ | $\mathbf{95.4 \pm 1.3}$ | $\mathbf{94.2 \pm 1.6}$ | $\mathbf{94.7 \pm 1.5}$ | $\mathbf{76.8 \pm 3.7}$ | $\mathbf{66.6 \pm 7.6}$ | $\mathbf{60.7 \pm 6.0}$ | $\mathbf{62.4 \pm 6.6}$ |

*Table 2.* Zero-shot classification performance across datasets and acquisition phases. MuHL and baselines are trained on a source dataset or phase and directly evaluated on unseen target datasets or phases without fine-tuning. Specifically, for PD, models are trained on PPMI and evaluated on TaoWu and Neurocon, while for AD, models are trained on ADNI-2 and evaluated on ADNI-1/3/GO.

| Method | ADNI-2→ADNI-1/3/GO | | | | PPMI→TaoWu | | | | PPMI→Neurocon | | | |
|---|---|---|---|---|---|---|---|---|---|---|---|---|
| | Acc ↑ | Pre ↑ | Rec ↑ | F1s ↑ | Acc ↑ | Pre ↑ | Rec ↑ | F1s ↑ | Acc ↑ | Pre ↑ | Rec ↑ | F1s ↑ |
| BrainGB | $42.9 \pm 3.2$ | $48.4 \pm 8.1$ | $36.5 \pm 2.4$ | $34.4 \pm 4.3$ | $53.0 \pm 1.2$ | $64.1 \pm 6.8$ | $53.0 \pm 1.2$ | $43.5 \pm 4.0$ | $62.9 \pm 6.4$ | $52.5 \pm 2.8$ | $51.6 \pm 7.3$ | $45.1 \pm 5.3$ |
| BrainNetTF | $43.5 \pm 8.4$ | $\underline{48.9 \pm 2.4}$ | $50.8 \pm 2.8$ | $44.8 \pm 1.6$ | $50.0 \pm 1.6$ | $46.0 \pm 3.9$ | $50.0 \pm 1.6$ | $46.2 \pm 4.6$ | $62.9 \pm 4.4$ | $62.7 \pm 8.1$ | $55.0 \pm 5.6$ | $50.3 \pm 8.7$ |
| ALTER | $43.8 \pm 3.0$ | $46.8 \pm 2.0$ | $54.6 \pm 6.4$ | $44.3 \pm 2.3$ | $54.5 \pm 1.9$ | $\underline{65.1 \pm 6.0}$ | $52.5 \pm 1.9$ | $45.3 \pm 4.4$ | $\underline{63.8 \pm 3.2}$ | $63.8 \pm 7.6$ | $55.3 \pm 4.4$ | $\underline{51.7 \pm 8.0}$ |
| BQN | $\underline{52.3 \pm 2.9}$ | $39.5 \pm 6.3$ | $32.8 \pm 7.7$ | $39.0 \pm 7.6$ | $\underline{57.0 \pm 4.0}$ | $56.3 \pm 4.6$ | $\underline{55.0 \pm 4.0}$ | $\underline{49.5 \pm 3.0}$ | $55.1 \pm 3.7$ | $47.8 \pm 3.2$ | $48.3 \pm 2.9$ | $46.8 \pm 2.9$ |
| dwHGCN | $52.8 \pm 2.9$ | $48.0 \pm 2.7$ | $\underline{57.0 \pm 2.8}$ | $\underline{48.8 \pm 3.1}$ | $50.5 \pm 2.9$ | $45.1 \pm 5.2$ | $50.5 \pm 2.9$ | $36.0 \pm 4.5$ | $60.9 \pm 2.7$ | $59.3 \pm 3.4$ | $53.6 \pm 3.7$ | $46.1 \pm 7.0$ |
| HyBRiD | $47.6 \pm 5.1$ | $46.7 \pm 4.6$ | $53.3 \pm 2.9$ | $44.0 \pm 5.2$ | $56.2 \pm 3.1$ | $53.8 \pm 3.5$ | $49.8 \pm 4.7$ | $44.7 \pm 3.8$ | $63.4 \pm 2.4$ | $51.7 \pm 2.5$ | $50.0 \pm 4.2$ | $48.8 \pm 3.6$ |
| MuHL | $\mathbf{56.0 \pm 1.6}$ | $\mathbf{52.1 \pm 6.1}$ | $\mathbf{58.4 \pm 8.3}$ | $\mathbf{54.8 \pm 7.7}$ | $\mathbf{60.5 \pm 1.6}$ | $\mathbf{72.8 \pm 2.4}$ | $\mathbf{60.5 \pm 1.6}$ | $\mathbf{58.9 \pm 3.0}$ | $\mathbf{65.9 \pm 3.4}$ | $\mathbf{68.1 \pm 3.3}$ | $\mathbf{57.3 \pm 3.2}$ | $\mathbf{53.7 \pm 3.6}$ |

**Baselines.** We categorized the baselines into two groups; 1) Graph-based models such as GCN (Kipf & Welling, 2016), GAT (Veličković et al., 2018), GCNII (Chen et al., 2020b), BrainGNN (Li et al., 2021), IBGNN (Cui et al., 2022b), BrainGB (Cui et al., 2022a), SGCN (Zhou et al., 2024), BrainNetTF (Kan et al., 2022), ALTER (Yu et al., 2024), BioBGT (Peng et al., 2025), and BQN (Yang et al., 2025) and 2) Hypergraph-based models such as HGNN (Feng et al., 2019), HNHN (Dong et al., 2020), UniGC-NII (Huang & Yang, 2021), HyperDrop, (Jo et al., 2021), dwHGCN (Wang et al., 2023), HyperGT (Liu et al., 2024), HyBRiD (Qiu et al., 2024), and DHHNN (Mei et al., 2025). Here, BrainGNN, IBGNN, SGCN, BrainGB, BrainNetTF, BioBGT, BQN, dwHGCN, and HyBRiD were specifically proposed to analyze the brain network.

**Setup.** We designed a 5-way classification for the ADNI dataset using multiple imaging biomarkers, and a 3-way classification for the PPMI dataset based on accumulated BOLD signals, both aiming to distinguish stages of neurodegenerative disease. For unbiased and stable results, 5-fold *cross validation* was used, and accuracy (Acc), precision (Pre), recall (Rec) and F1-score (F1s) in their mean were computed for evaluation. We set up and trained each baseline to achieve feasible outcomes for fair comparisons. More details are given in the Appendix C.

**Results.** The classification performances from MuHL and 19 baselines are reported in Table 1. As in Table 1, MuHL consistently outperformed all baselines on the ADNI dataset, improving accuracy by 2.4%p, precision by 2.1%p, recall by 4.6%p, and F1 score by 3.8%p over the second-best method. In addition to its superior performance, MuHL exhibits stability, as evidenced by low standard deviations across 5-fold evaluations. To further assess effectiveness in multi-modal analysis and evaluate the contribution of each biomarker from the ADNI, we conducted additional experiments under various scenarios, which are in the Appendix D.

For the PPMI, MuHL also brings improvements with average gain of 3.9%p in accuracy, 3.4%p in precision, 4.9%p in recall, and 6.0%p in F1-score, outperforming all baselines in both precision and recall where most methods struggled. Notably, the substantial gain in recall is particularly important in medical applications, as it stems from high-order interactions often overlooked by conventional methods. In addition, MuHL outperformed recent approaches for brain network analysis, demonstrating its robustness and effectiveness in refining hypergraph structures based on multi-resolution filtered graph signals for brain network classification.

**Zero-Shot Learning.** To evaluate the generalizability of MuHL under heterogeneous distribution shifts, Table 2 re-

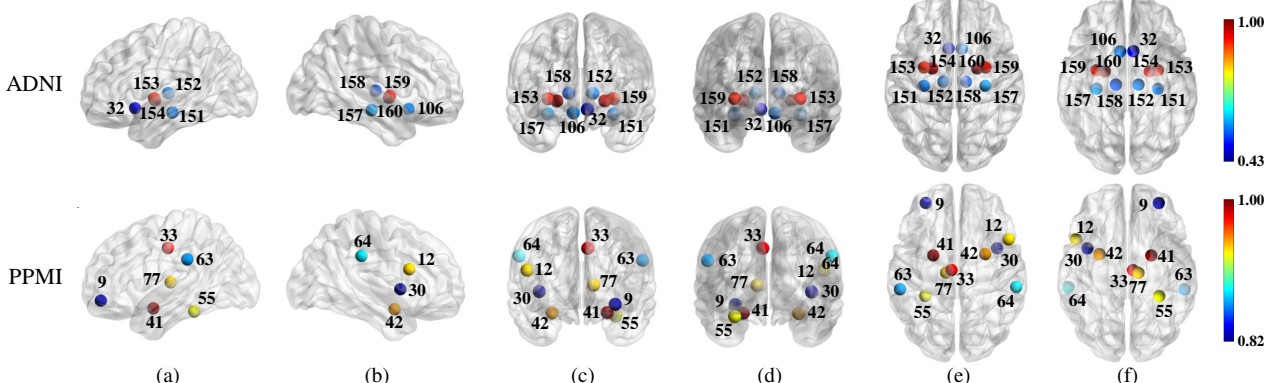

*Figure 3.* Visualization of top-10 ROIs with the highest relative hyperedge importance on the ADNI (top) and PPMI (bottom) datasets. (a)/(b): outer views of left/right hemisphere, (c)/(d): front/rear views, (e)/(f): top/bottom views. Node color reflects the relative contribution of each ROI based on aggregated hyperedge weights, with node indices corresponding to the Destrieux atlas (Destrieux et al., 2010) for the ADNI dataset and the AAL atlas (Tzourio-Mazoyer et al., 2002) for the PPMI dataset.

*Table 3.* ROIs with the 10 highest relative importance of aggregated hyperedge activation (Act.) for classification. (L) and (R) denote the left and right hemispheres, respectively.

| ADNI | | | | | | PPMI | | | | | |
|---|---|---|---|---|---|---|---|---|---|---|---|
| Idx | ROI | Act. | Idx | ROI | Act. | Idx | ROI | Act. | Idx | ROI | Act. |
| 154 | (L) Globus.Pallidus | 1.000 | 160 | (R) Globus.Pallidus | 0.963 | 41 | (L) Amygdala | 1.000 | 33 | (L) G.Cingulum.Mid | 0.974 |
| 159 | (R) Putamen | 0.944 | 153 | (L) Putamen | 0.941 | 42 | (R) Amygdala | 0.948 | 77 | (L) Thalamus | 0.939 |
| 157 | (R) Hippocampus | 0.551 | 151 | (L) Hippocampus | 0.532 | 12 | (R) G.Frontal.Inf.Oper | 0.939 | 55 | (L) G.Fusiform | 0.929 |
| 106 | (R) G.Subcallosal.Area | 0.529 | 152 | (L) Thalamus | 0.528 | 64 | (R) G.Supramarginal | 0.888 | 63 | (L) G.Supramarginal | 0.870 |
| 158 | (R) Thalamus | 0.499 | 32 | (L) G.Subcallosal.Area | 0.432 | 9 | (L) G.Frontal.Mid.Orb | 0.829 | 30 | (R) Insula | 0.829 |

ports zero-shot classification performance across both acquisition phases and independently collected datasets. For AD, models are trained on ADNI-2 and evaluated on unseen target phases (ADNI-1/3/GO), where MuHL consistently outperforms baselines, achieving improvements of up to 3.7%p in accuracy and 6.0%p in F1-score over the second-best method despite substantial phase-wise differences. For PD, MuHL is trained on PPMI and evaluated on the TaoWu and Neurocon cohorts from Xu et al. (2023) without fine-tuning, and MuHL surpasses the second-best method by 7.7%p in precision and 5.5%p in recall on TaoWu, and by 4.3%p in precision and 2.0%p in recall on Neurocon. These results demonstrate that MuHL captures transferable disease-related structural representations that remain robust across cross-phase and cross-dataset shifts, enabled by its multi-scale and adaptive high-order structural modeling.

## 6. Discussions on the Learned Hypergraph

**Identifying Key ROIs via Hyperedge Activations.** Understanding the importance of ROIs in brain networks is essential for analyzing disease progression, as these regions play a crucial role in characterizing disease-related alterations. The aggregated hyperedge activations (i.e., degree) for each ROI serve as indicators of its network significance. As shown in Fig. 3, ROIs with high cumulative activations act as network hubs, facilitating information flow and increasing sensitivity to pathological changes in both AD and PD. Identifying these key ROIs highlights disease-relevant regions most affected by neurodegeneration, enhancing both

model interpretability and clinical relevance.

Top-10 ROIs with highest importance of cumulative hyperedge activations, listed in Table 3, are primarily subcortical regions in both ADNI and PPMI datasets, highlighting their role in disease-stage classification and brain network communication. In ADNI, 4 ROIs (*globus pallidus*, *putamen*, *thalamus*, and *hippocampus*) appear as left/right pairs, reflecting a symmetrical pattern across hemispheres. These subcortical regions, serving as key hubs for sensory, motor, and cognitive integration (de Jong et al., 2008; Rao et al., 2022), consistently demonstrate their significance across multiple modalities, revealing their crucial role in neurodegenerative progression. Similarly in PPMI, the *amygdala* and *thalamus* stand out as major subcortical contributors, while the *amygdala* and *supramarginal gyrus* form symmetric left/right pairs, suggesting a bilateral pattern linked to PD-specific alterations in emotion, motor, and cognitive regulation (Harding et al., 2002b; Yoshimura et al., 2005). These findings demonstrate the ability of MuHL to identify disease-relevant and symmetric ROI groups, particularly in subcortical structures, across both AD and PD spectra.

**Interpreting the Most Salient Hyperedge.** To further investigate the sub-structure of the learned hyperedges, we analyzed the most prominent hyperedge identified by aggregating the activation values of all nodes within each hyperedge. In Fig. 4, a subset of ROIs from the ADNI dataset, e.g., *hippocampus* and *putamen*, are not only co-activated within the same hyperedge but also overlap with those highlighted in Fig. 3 as prominent network hubs, indicating that

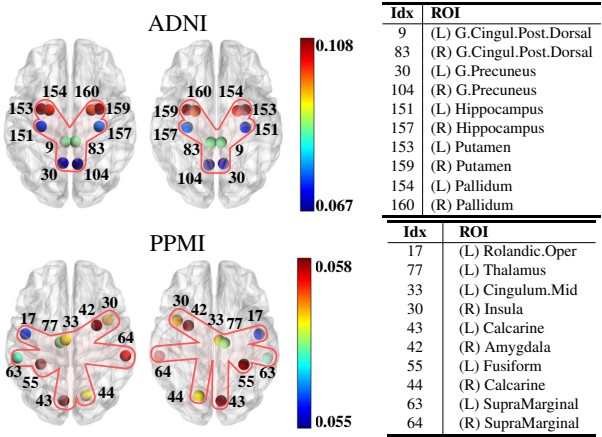

Figure 4. Left: Visualization of the top-10 ROIs associated with the most important hyperedge for top/bottom views in the ADNI and PPMI studies, based on the sum of node activations for each hyperedge. Node color indicates activation value (i.e., weight) per node within the hyperedge. Right: Corresponding ROI labels.

these regions are functionally interrelated across both global and localized contexts (Lai et al., 2017; Yamashita et al., 2019). Notably, five of the top ROIs in this representative hyperedge appear as symmetric left–right pairs, suggesting a bilateral organization in the disease-associated brain network. For PPMI dataset, the most salient hyperedge was characterized by integrating node activations across hyperedges. As shown in the right of Fig. 4, top-10 ROIs include important regions for PD such as *thalamus* and *amygdala*, which are critical for executive functioning and emotional processing (Harding et al., 2002a; Dirkx & Bologna, 2022). These observations suggest that MuHL captures both individual regional importance and inter-regional relationships through hyperedges, supporting functional connectivity and their role in neurodegenerative pathology.

## 7. Effectiveness of Model Components

**Hyper-parameter Studies.** To explore sensitivity of our model to hyper-parameters, we examine the impact of the number of scales $J$, the number of hyperedges $M$, the maximum number of neighboring hyperedges $\eta$, and the embedding dimension $d_h$ in the hypergraph structure learning. Default values were set as $J=3$, $M=16$, $\eta=3$, and $d_h=16$ with the best performance, and we varied each factor independently. As seen in Fig. 5, for the $M$ (i.e. orange), accuracy increased up to 16 but declined beyond 20, likely due to noisy or redundant hyperedges. Regarding the $d_h$ (i.e. green), performance improved up to 16 and then plateaued, suggesting that while increasing $d_h$ initially enhances representational capacity, further growth may lead to overfitting.

**Ablation Studies.** To assess the effect of each component in MuHL, we conduct an ablation study on the ADNI and PPMI datasets, as summarized in Table 4. MuHL integrates three key components: adaptive multi-scale feature

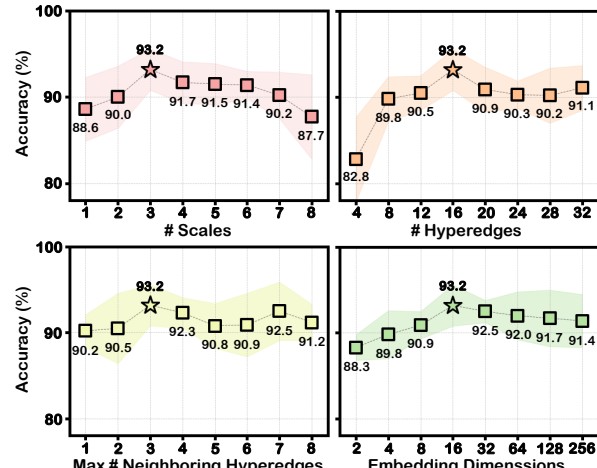

Figure 5. Sensitivity studies of crucial hyperparameters in MuHL on the ADNI dataset. Boxes denote the mean, with shading indicating standard deviation across folds.

filtering (MSF), hypergraph structure learning (HSL), and multi-scale transformer (MST). Removing any of these components leads to a clear performance drop, confirming their complementary contributions to modeling high-order brain structure. Specifically, MSF provides scale-aware representations through graph wavelet filtering, HSL learns data-driven hyperedges that reflect regional groupings, and MST enables context-aware information aggregation across resolutions. Among the variants, removing HSL results in the most significant degradation, highlighting the importance of flexible hyperedge modeling in capturing the irregular topology of brain networks. These results demonstrate that all three modules are crucial in combination, and jointly contribute to the strong predictive performance of MuHL.

Table 4. Ablation study of MuHL components on both datasets.

| Component | | | ADNI | | | PPMI | | |
|---|---|---|---|---|---|---|---|---|
| MSF | HSL | MST | Acc | Pre | Rec | Acc | Pre | Rec |
| ✗ | ✓ | ✓ | $90.0_{\pm2.8}$ | $91.4_{\pm3.7}$ | $89.9_{\pm4.4}$ | $72.9_{\pm4.6}$ | $62.1_{\pm9.8}$ | $55.7_{\pm5.7}$ |
| ✓ | ✗ | ✓ | $76.8_{\pm1.9}$ | $76.9_{\pm7.0}$ | $72.4_{\pm6.6}$ | $67.4_{\pm5.9}$ | $45.5_{\pm9.4}$ | $41.6_{\pm4.1}$ |
| ✓ | ✓ | ✗ | $86.9_{\pm3.4}$ | $90.4_{\pm2.4}$ | $84.4_{\pm5.0}$ | $64.1_{\pm1.8}$ | $53.0_{\pm8.6}$ | $44.5_{\pm6.9}$ |
| ✓ | ✓ | ✓ | $\mathbf{93.2_{\pm2.4}}$ | $\mathbf{95.4_{\pm1.3}}$ | $\mathbf{94.2_{\pm1.6}}$ | $\mathbf{76.8_{\pm3.7}}$ | $\mathbf{66.6_{\pm7.6}}$ | $\mathbf{60.7_{\pm6.0}}$ |

## 8. Conclusion

In this work, we proposed MuHL, a novel hypergraph framework to adaptively model high-order relationships in brain networks. By leveraging multi-resolution graph representations and dynamically constructing hyperedges, our method effectively captures complex inter-regional dependencies beyond pairwise connections. Extensive experiments on the ADNI and PPMI datasets show that MuHL demonstrates superiority as evidenced by improved performance in AD- and PD-specific stages classification. Furthermore, the trained hypergraph identifies hub ROIs linked to disease progression across both AD and PD, showcasing the potential of adaptive hypergraph learning as a powerful tool for analyzing diverse neurodegenerative disorders.

## Acknowledgements

This research was supported by RS-2025-02216257 (70%), RS-2026-25494850 (20%), and RS-2019-II1091906 (AI Graduate Program at POSTECH, 10%).

## Impact Statement

This work targets stage classification of neurodegenerative disorders, where improved sensitivity and stability may support earlier detection and patient stratification. The framework identifies salient ROIs and group-wise interactions through learned hyperedges, providing interpretable signals that complement conventional imaging biomarkers. However, the performance may be less reliable for underrepresented disease groups due to dataset imbalance, potentially leading to biased predictions. In addition, the identified ROIs and hyperedges should not be interpreted as causal biomarkers, but rather as correlational patterns learned from data. The method is intended to support, not replace, clinical decision-making, and should be used with established diagnostic criteria with appropriate human oversight.

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

This material presents the supplementary paper from the main manuscript due to the space limitations. In Section A, the proof of the propositions on the foundation for our hypergraph structure learning is provided. In Section B, a detailed description of the ADNI and PPMI datasets used in our experiments is provided. Section C outlines the full implementation details and hyperparameters of MuHL and baseline methods for AD classification. Section D presents tri-modal leave-one-out ablations and dual-modality fusion results, quantifying marginal effect of each biomarker and the synergy of pairwise combinations. We additionally conducted experiments on other brain disorder dataset to further validate the generality of our approach. In Section E, we discuss additional behavior of the optimized hypergraph. In Section F, we additionally conduct regression experiments to elucidate the relationship between structural/functional brain alterations and clinical scores. Finally, we conducted additional ablation studies in Section G, detailed model complexity is explained in Section H, and we also discuss our framework in Section I.

## A. Proof of Propositions

**Proposition 1.** *Given a graph signal $X$ and a hyperedge projector $\Phi$, computing incidence matrix $H_s$ is equivalent to projecting the graph wavelet representation of $X$, i.e., $W_X(s)$, with $\Phi$.*

*Proof.* Let $X \in \mathbb{R}^{N \times D}$ be the node feature, where $N$ is the number of nodes and $D$ is the feature dimension of $X$. From Hammond et al. (2011), the graph signal $X$ is decomposed into various levels of granularity in the spectral space for multi-resolution graph analysis. To project the signals into the spectral space, the spectral graph wavelet basis at scale $s$ is defined as

$$\psi_s = U g(s\Lambda) U^T, \tag{9}$$

where $U$ and $\Lambda$ are the eigenvectors and eigenvalues of the normalized Laplacian $\hat{\mathcal{L}}$, and $g(\cdot)$ is the wavelet kernel. Since $U$ is orthonormal and $g(s\Lambda)$ is diagonal, $\psi_s^T$ can be replaced by $\psi_s$. Now, the wavelet coefficients of the node signals at scale $s$ is computed as

$$\begin{aligned} W_X(s) &= \langle \psi_s, X \rangle \\ &= \psi_s X. \end{aligned} \tag{10}$$

Using (9), the filtered node representation in the graph space at the scale $s$ is defined as

$$\begin{aligned} X_s &= \langle \psi_s, W_X(s) \rangle \\ &= \psi_s W_X(s) \in \mathbb{R}^{N \times D} \\ &= \psi_s \psi_s X \\ &= \psi_s U g(s\Lambda) U^T X \\ &= U g(s\Lambda) U^T U g(s\Lambda) U^T X \\ &= U g^2(s\Lambda) U^T X. \end{aligned} \tag{11}$$

Let $\Phi \in \mathbb{R}^{D \times M}$ be a hyperedge projector with $M$ hyperedges that maps node features into the hyperedge space. Using $X_s$ and $\Phi$, the scale-dependent incidence matrix is constructed as

$$H_s = X_s \Phi. \tag{12}$$

Substituting Eq. (12) into Eq. (11) results in

$$\begin{aligned} H_s &= X_s \Phi \in \mathbb{R}^{N \times M} \\ &= U g^2(s\Lambda) U^T X \Phi \\ &= U g(s\Lambda)^T g(s\Lambda) U^T X \Phi \\ &= U g(s\Lambda)^T U^T U g(s\Lambda) U^T X \Phi \\ &= U g(s\Lambda)^T U^T \psi_s X \Phi \\ &= U g(s\Lambda)^T U^T W_X(s) \Phi \\ &= U g(s\Lambda) U^T W_X(s) \Phi \\ &= \psi_s W_X(s) \Phi. \end{aligned} \tag{13}$$

Thus, Eq. (14) shows that the incidence matrix $H_s$ is ultimately constructed from the wavelet coefficients $W_X(s)$ combined with the hyperedge projector $\Phi$, providing a spectral formulation of node-to-hyperedge relations. $\qquad\square$

In practice, we apply a linear embedding and use $\bar{X}_s = X_s\,\Theta$ due to the low input dimensionality, where $\Theta \in \mathbb{R}^{D \times d_h}$. All subsequent computations are the same as $H_s = \psi_s W_{\bar{X}_s}\Phi$. This parameterization ensures that the statement and proof of Proposition 1 remain unchanged. Finally, $H_s$ is normalized to obtain $\tilde{H}_s$ and truncated via TopK to produce $\bar{H}_s$, as in Eq. (2) and Eq. (3).

**Proposition 2.** *In $H_s$, there exists at least one hyperedge $e_m^*$ such that the number of nodes within $e_m^*$, for which $e_m^*$ attains the highest weight, monotonically increases with respect to s, and this number converges to $N$ as $s \to \infty$ , i.e., $\lim_{s\to\infty} |\{v_n : e_m^* = \mathrm{argmax}_{e_m} H_s(v_n, e_m)\}| \approx N$.*

*Proof.* We prove the Proposition 2 by considering separately the cases of low-pass and band-pass kernels. Although their limiting behaviors differ, both cases exhibit the same scale-dependent trend that hyperedges expand as the scale increases.

**(i) Low-pass kernels.** Suppose that $g(\cdot)$ is a low-pass kernel with $g(0)=1$ and $g(s\lambda) \to 0$ as $s \to \infty$ for all $\lambda > 0$. In this case, as the scale grows, all high-frequency coefficients corresponding to eigenvalues $\lambda_k$ with $k \geq 2$ vanish. Consequently, from Eq. (11), the filtered node features converge to the projection on the the first eigenvector as

$$
\begin{aligned}
X_s &= \sum_{i=1}^{N} g^2(s\lambda_i) u_i u_i^T X \\
&= \underbrace{g^2(s\lambda_1)}_{=1} u_1 u_1^T X + \underbrace{g^2(s\lambda_2) u_2 u_2^T X + \cdots + g^2(s\lambda_N) u_N u_N^T X}_{\approx 0} \\
&\approx u_1(u_1^T X).
\end{aligned}
\tag{14}
$$

This shows that all node features eventually collapse onto the one-dimensional subspace spanned by $u_1$, so that the variability across nodes disappears and only the common low-frequency mode remains. When Eq. (14) is substituted into the Eq. (12), which is computed as

$$
\begin{aligned}
H_s &= X_s\Phi \\
&\approx u_1(u_1^T X)\Phi.
\end{aligned}
\tag{15}
$$

Here, each row of $H_s$ can be written as

$$
H_s(v_n, :) \approx u_{1,v_n}(u_1^T X)\Phi,
\tag{16}
$$

where $u_{1,v_n}$ denotes the $n$-th entry of the first eigenvector. Since all $u_{1,v_n}$ are positive and nearly uniform across nodes, the weight scores from $H_s(v_n, :)$ differ only by a scalar multiple. Thus, the relative ordering of hyperedge scores is the same for every node. The argmax over columns (i.e, hyperedges) selects the same hyperedge projection $e_m^*$ for all $n$. As a result, the degree of this hyperedge increases monotonically with the scale and converges to the total number of nodes as

$$
\lim_{s\to\infty} |\{v_n : e_m^* = \mathrm{argmax}_{e_m} H_s(v_n, e_m)\}| = N.
\tag{17}
$$

**(ii) Band-pass kernels.** Consider a band-pass kernel with $g(0)=0$ and maximum response at positive eigenvalues. The peak response shifts toward the smallest nonzero eigenvalues at the scale $s$ increases. Thus, there exists a small eigenvalue interval $\mathcal{I}=[\lambda_2, \lambda_n]$ around $\lambda_2$ such that only components with $\lambda_k \in \mathcal{I}$ remain significant, and we denote the corresponding index set as $\mathcal{K}^*=\{k : \lambda_k \in \mathcal{I}\}$. In this interval, the filtered representations can be approximated as

$$
\begin{aligned}
X_s &= \sum_{i=1}^{N} g^2(s\lambda_i) u_i u_i^T X \\
&= \underbrace{g^2(s\lambda_1)}_{=0} u_1 u_1^T X + \underbrace{\cdots + g^2(s\lambda_n) u_n u_n^T X}_{\lambda_k \in \mathcal{I}} + \cdots + g^2(s\lambda_N) u_N u_N^T X \\
&\approx \sum_{k\in\mathcal{K}^*} g^2(s\lambda_k) u_k u_k^T X,
\end{aligned}
\tag{18}
$$

so that the node representations concentrate in the low-frequency subspace spanned by the eigenvectors $\{u_k : k \in \mathcal{K}^*\}$. Contributions outside this subspace decay relative to the dominant components as $s$ increases. Plugging Eq. (18) into the Eq. (12), as

$$H_s = X_s \Phi$$
$$\approx \left( \sum_{k \in \mathcal{K}^*} u_i u_i^T X \right) \Phi \tag{19}$$

In this form, the variation among nodes is confined to a shared low-dimensional subspace. If one hyperedge projection $e_m^*$ is better aligned with this subspace than all others, then its score advantage becomes increasingly dominant as the residual vanishes. Consequently, the number of nodes for which $e_m^*$ yields the highest weight grows monotonically with the scale. In the asymptotic regime, this number converges to the size of the node set aligned with the dominant low-frequency subspace, which can be close to the full node set as

$$\lim_{s \to \infty} |\{v_n : e_m^* = \text{argmax}_{e_m} H_s(v_n, e_m)\}| \approx N. \tag{20}$$

These two cases demonstrate that low-pass kernels drive all nodes into the same $u_1$ direction so that a specific hyperedge eventually covers all nodes, while band-pass kernels concentrate the features in a fixed low-frequency subspace on certain scale intervals so that the best-aligned hyperedge expands its degree within those intervals. In either case, increasing the wavelet scale smooths node features and enlarges hyperedges, consistent with the patterns observed in Fig. (1). $\square$

## B. Dataset Description

To understand our experiments, we describe the datasets and preprocessing steps of our experiments in detail. We used two neurodegenerative brain network benchmarks: the Alzheimer's Disease Neuroimaging Initiative (ADNI) and the Parkinson's Progression Markers Initiative (PPMI).

**ADNI dataset.** The ADNI study (Mueller et al., 2005) is a publicly available dataset designed for Alzheimer's Disease (AD) research, providing multi-modal imaging and biomarker data across various stages of cognitive decline. The structural brain graphs in the ADNI dataset are constructed through a multi-step imaging pipeline. The image preprocessing steps include 1) performing skull stripping and tissue segmentation on T1-MRI; 2) performing brain parcellation using the Destrieux atlas; 3) obtaining diffusion tensor information from DWI data; 4) constructing cortical surface using free-surfer; 5) applying probabilistic tractography to construct structural brain network; 6) measuring region-wise imaging features such as Standard Uptake Value Ratio (SUVR) of tau protein from Tau-PET (TAU), $\beta$-amyloid protein from Amyloid-PET (AMY), metabolism level from FDG-PET (FDG) and cortical thickness from MRI (CT), where the cerebellum was used as the reference for the SUVR normalization.

Following the clinical outcomes, the diagnostic labels for each subject were defined as Control (CN), Significant Memory Concern (SMC), Early-stage Mild Cognitive Impairment (EMCI), Late-stage Mild Cognitive Impairment (LMCI), and Alzheimer's Disease (AD). The demographics of the ADNI dataset is summarized in Table 5. It includes a total of 650 subjects, with 226 CN, 131 SMC, 217 EMCI, 64 LMCI, and 12 AD cases. Table 5 contains group-wise details including gender, age, and multiple AD-related clinical symptoms such as CDR-SOB, ADAS11, ADAS13, and MMSE. The average age within these groups ranges from 68.58 to 73.46 years, with standard deviations between 4.76 and 10.34 years. Gender distribution varies across groups, with CN comprising 114 males and 112 females, while AD includes 9 males and 3 females.

**PPMI dataset.** The PPMI dataset is designed to identify biological markers related to the risk, onset, and progression of Parkinson's disease (PD), a neurodegenerative disorder primarily affecting movement. For brain network extraction, resting-state fMRI is parcellated into 116 regions based on the AAL atlas to obtain Blood-Oxygen-Level-Dependent (BOLD)

*Table 5.* Demographics of the ADNI dataset.

| Category | CN | SMC | EMCI | LMCI | AD |
|---|---|---|---|---|---|
| # of subjects | 226 | 131 | 217 | 64 | 12 |
| Gender (M / F) | 114 / 112 | 42 / 89 | 121 / 96 | 43 / 21 | 9 / 3 |
| Age (Mean ± Std) | 71.27 ± 5.92 | 71.42 ± 4.76 | 68.58 ± 7.21 | 69.66 ± 5.25 | 73.46 ± 10.34 |
| CDR-SOB (Mean ± Std) | 0.02 ± 0.12 | 0.16 ± 0.40 | 1.17 ± 0.89 | 1.27 ± 0.98 | 3.54 ± 2.14 |
| ADAS11 (Mean ± Std) | 4.70 ± 2.51 | 4.39 ± 2.45 | 7.24 ± 3.30 | 8.99 ± 3.79 | 16.74 ± 4.66 |
| ADAS13 (Mean ± Std) | 7.14 ± 3.87 | 6.52 ± 3.72 | 11.28 ± 5.16 | 14.76 ± 5.85 | 27.03 ± 5.22 |
| MMSE (Mean ± Std) | 29.07 ± 1.24 | 29.08 ± 1.13 | 28.48 ± 1.58 | 28.23 ± 1.64 | 22.75 ± 4.05 |

signals, which are then used to compute the connectivity matrix; this preprocessing was conducted by (Marek et al., 2011). Node features are extracted from BOLD signals capturing changes in blood flow, where the mean BOLD signal per region is used to account for different lengths.

Based on clinical outcomes, disease labels were assigned to each subject as CN, Prodromal, or PD. Table 6 summarizes the demographics of the PPMI dataset, which consists of 181 subjects: 15 CN, 53 Prodromal, and 113 PD. The average ages of these groups range from 62.03 to 64.28 years, with standard deviations between 7.77 and 10.03 years.

*Table 6.* Demographics of the PPMI dataset.

| Category | Control | Prodromal | PD |
|---|---|---|---|
| # of subjects | 15 | 53 | 113 |
| Gender (M / F) | 12 / 3 | 30 / 23 | 77 / 36 |
| Age (Mean $\pm$ Std) | $62.03 \pm 10.03$ | $64.00 \pm 9.54$ | $64.28 \pm 7.77$ |

## C. Implementation Details

**Experimental Settings.** Our method and all baselines were implemented using PyTorch and trained on a single NVIDIA RTX 6000 Ada Generation. All experiments were conducted with a fixed random seed to ensure reproducibility. We used Adam optimizer and trained each model for 1000 epochs. Due to data sparsity, we employed a 5-fold cross-validation strategy with fixed hyperparameters to evaluate the model's performance. A grid search was conducted over the following common hyperparameter ranges: the number of hidden units in $\{2, 4, 8, 16, 32, 64\}$, dropout rates in $\{0.2, 0.5, 0.8\}$, learning rates in $\{10^{-1}, 10^{-2}, 10^{-3}, 10^{-4}\}$, weight decay in $\{5e\text{-}2, 5e\text{-}3, 5e\text{-}4\}$. The batch size was set to include all samples in each dataset. We set up and trained each baseline to achieve feasible outcomes for fair comparisons. For this, we set the number of hidden units to 16 for the ADNI dataset and 8 for the PPMI dataset across all baselines, and standardized key hyperparameters, with a dropout rate of 0.5, a learning rate of 1e-3, and a weight decay of 5e-4.

**Our Details.** For multi-scale analysis from MuHL, we set two wavelet kernels: one is a low-pass filter $g(s)=e^{-sx}$ that captures node signals in the low-frequency and the other $g(s)$ based on (Hammond et al., 2011) is a band-pass filter which is 0 at the origin. The combined effect of low-pass and band-pass filters is obtained by applying a low-pass kernel at one scale and band-pass filters at the remaining $J-1$ scales. This approach enables selective extraction of spectral components across multiple resolutions. Here, the number of scales $J$ was empirically determined by searching over $\{1, 2, 3, 4, 5, 6, 7, 8\}$, and $J=3$ was finally selected for the ADNI dataset, while $J=2$ was chosen for the PPMI dataset, as they respectively yielded the best performance. The initial scale for the low-pass filter was set to 2, and the $J-1$ scales for the band-pass filter were initialized within a range $(0,2]$. These scale values are treated as learnable parameters and are adaptively updated during training to appropriate positive values. The number of hyperedges $M$ was empirically determined by finding over $\{4, 8, 12, 16, 20, 24, 28, 32\}$, and $M=16$ for ADNI and $M=12$ for PPMI were selected to make the best performance.

We stacked $Z=2$ hypergraph convolution layers with rectified linear unit (ReLU) as the activation function $\sigma$ and $P=1$ multi-scale attention layer with softmax function at the output to predict the graph class. Unlike previous works that assign learnable weights to hyperedges via $W_e$, we fix $W_e=1$ and instead focus on learning the incidence matrix $H$ directly. This design enables more flexible and fine-grained modeling of node–hyperedge relationships, as it allows the model to selectively control hyperedge connectivity patterns rather than merely adjusting their scalar strengths. Also, we set the loss weight $\alpha=1$ to ensure a balanced contribution between the cross-entropy and the scale-based regularization in the objective function.

**Baseline-specific Details.** To ensure both fairness and reproducibility, we utilized the official open-source implementation for all baseline methods. In addition to the shared grid search, we performed further tuning of key hyperparameters specific to each model to better adapt them to our domain-specific datasets.

- **GAT** (Veličković et al., 2018): We tuned the number of transformer head $\{2, 4, 8\}$.

- **BrainGNN** (Li et al., 2021): We adjusted the number of ROI communities $\{2, 4, 6, 8\}$ and the pooling ratio for R-pool layer $\{0.1, 0.2, 0.3, 0.4, 0.5\}$.

- **BrainGB** (Cui et al., 2022a): We selected the number of buckets $T \in \{5, 10, 15, 20\}$ to cluster region connections with similar strengths.

- **SGCN** (Zhou et al., 2024): We tuned hyper-parameters $\lambda_1 \in \{1, 0.1, 0.01\}$, $\lambda_2 \in \{1, 0.1, 0.01\}$, and $\lambda_3 \in \{1, 0.1, 0.01\}$ as penalty coefficients for different regularization.

*Table 7.* AD classification performance (CN/SMC/EMCI/LMCI/AD) on the ADNI dataset under various scenarios. The best results are shown in **bold**, and the second-best results are underlined. (†: Methods for brain network analysis)

| Category | Method | CT+AMY+FDG | | | | CT+AMY+TAU | | | |
|---|---|---|---|---|---|---|---|---|---|
| | | Acc ↑ | Pre ↑ | Rec ↑ | F1s ↑ | Acc ↑ | Pre ↑ | Rec ↑ | F1s ↑ |
| Graph based method | GCN | 84.9 ± 1.7 | 85.3 ± 7.6 | 80.3 ± 7.1 | 81.9 ± 7.5 | 85.5 ± 1.6 | 87.3 ± 6.3 | 81.7 ± 3.9 | 83.6 ± 4.5 |
| | GAT | 86.8 ± 2.3 | 89.4 ± 1.4 | 85.6 ± 2.5 | 86.9 ± 2.1 | 88.3 ± 1.8 | 91.3 ± 1.8 | 84.0 ± 2.3 | 86.4 ± 1.9 |
| | GCNII | 78.8 ± 4.0 | 86.9 ± 2.8 | 77.1 ± 6.3 | 79.9 ± 5.4 | 81.8 ± 4.3 | 87.1 ± 2.5 | 80.1 ± 7.4 | 85.2 ± 5.3 |
| | IBGNN† | 75.4 ± 5.1 | 81.2 ± 3.5 | 77.6 ± 6.1 | 78.3 ± 5.4 | 74.2 ± 4.4 | 78.7 ± 4.3 | 74.0 ± 3.5 | 75.2 ± 1.9 |
| | BrainGB† | 78.8 ± 2.2 | 84.0 ± 3.2 | 77.7 ± 3.2 | 79.5 ± 2.7 | 80.5 ± 2.5 | 86.1 ± 2.9 | 77.3 ± 3.5 | 80.2 ± 2.9 |
| | SGCN† | 81.2 ± 1.8 | 85.2 ± 3.2 | 81.8 ± 2.3 | 82.8 ± 1.9 | 82.2 ± 1.1 | 86.5 ± 3.0 | 83.7 ± 2.1 | 84.3 ± 1.8 |
| | BrainNetTF† | 89.2 ± 1.8 | 91.4 ± 2.1 | 86.6 ± 7.1 | 87.8 ± 5.5 | 86.8 ± 2.0 | 90.1 ± 1.7 | 84.5 ± 6.1 | 86.2 ± 4.8 |
| | ALTER† | 89.1 ± 2.1 | 87.6 ± 8.1 | 85.6 ± 8.9 | 86.0 ± 9.4 | 87.5 ± 2.3 | 90.6 ± 1.4 | 85.1 ± 5.9 | 86.7 ± 4.3 |
| | BioBGT† | 79.2 ± 2.3 | 78.5 ± 3.2 | 77.3 ± 5.8 | 77.1 ± 4.3 | 77.8 ± 3.1 | 76.9 ± 4.4 | 76.3 ± 2.1 | 76.7 ± 4.2 |
| Hypergraph based model | HGNN | 84.3 ± 2.5 | 88.3 ± 1.9 | 83.3 ± 4.9 | 84.9 ± 3.5 | 80.2 ± 1.1 | 82.1 ± 7.2 | 76.0 ± 6.7 | 78.1 ± 6.6 |
| | HNHN | 86.8 ± 1.6 | 89.2 ± 1.7 | 85.9 ± 5.0 | 86.8 ± 4.0 | 84.3 ± 2.0 | 86.9 ± 1.8 | 84.8 ± 4.9 | 85.4 ± 3.7 |
| | UniGCNII | 88.6 ± 1.9 | 89.4 ± 2.0 | 86.6 ± 4.7 | 77.5 ± 3.7 | 88.3 ± 2.7 | 87.2 ± 4.1 | 86.3 ± 4.9 | 86.5 ± 4.4 |
| | dwHGCN† | 88.8 ± 2.7 | 91.4 ± 2.7 | 86.9 ± 3.7 | 88.5 ± 3.1 | 85.5 ± 1.5 | 88.6 ± 2.3 | 85.6 ± 4.4 | 86.5 ± 2.9 |
| | HyperGT | 78.3 ± 2.9 | 82.2 ± 9.2 | 69.4 ± 7.4 | 73.4 ± 7.3 | 76.6 ± 3.5 | 74.2 ± 9.5 | 63.0 ± 5.2 | 65.8 ± 5.7 |
| | HyBRiD† | 86.5 ± 1.5 | 84.7 ± 3.8 | 86.8 ± 4.6 | 85.2 ± 3.8 | 79.2 ± 4.2 | 80.0 ± 6.3 | 79.8 ± 5.9 | 79.4 ± 6.0 |
| | MuHL | **91.0 ± 2.2** | **92.9 ± 1.9** | **90.1 ± 3.3** | **91.0 ± 3.3** | **88.9 ± 1.9** | **92.9 ± 2.0** | **87.4 ± 5.0** | **89.4 ± 3.9** |

| Category | Method | CT+FDG+TAU | | | | AMY+FDG+TAU | | | |
|---|---|---|---|---|---|---|---|---|---|
| | | Acc ↑ | Pre ↑ | Rec ↑ | F1s ↑ | Acc ↑ | Pre ↑ | Rec ↑ | F1s ↑ |
| Graph based method | GCN | 83.5 ± 2.7 | 87.0 ± 4.7 | 81.0 ± 4.8 | 82.5 ± 3.7 | 86.6 ± 2.7 | 85.9 ± 3.8 | 83.2 ± 5.8 | 83.4 ± 4.3 |
| | GAT | 87.4 ± 3.0 | 91.5 ± 1.8 | 86.2 ± 5.1 | 88.1 ± 3.9 | 88.9 ± 3.2 | 91.3 ± 3.5 | 86.6 ± 4.7 | 88.0 ± 3.8 |
| | GCNII | 81.1 ± 4.8 | 85.2 ± 7.6 | 77.0 ± 7.8 | 79.3 ± 7.5 | 84.2 ± 4.4 | 85.3 ± 7.6 | 81.2 ± 9.3 | 82.3 ± 8.2 |
| | IBGNN† | 82.9 ± 2.5 | 83.8 ± 3.5 | 79.1 ± 2.6 | 80.5 ± 2.7 | 81.7 ± 2.5 | 84.6 ± 5.2 | 80.4 ± 3.7 | 81.2 ± 3.6 |
| | BrainGB† | 81.8 ± 4.0 | 82.5 ± 7.9 | 81.1 ± 6.6 | 80.3 ± 6.7 | 82.8 ± 2.5 | 86.3 ± 4.0 | 82.7 ± 5.4 | 83.6 ± 4.1 |
| | SGCN† | 81.7 ± 2.1 | 87.9 ± 2.4 | 78.2 ± 4.7 | 81.3 ± 4.0 | 84.2 ± 2.1 | 89.2 ± 0.8 | 80.7 ± 4.8 | 83.5 ± 3.7 |
| | BrainNetTF† | 88.6 ± 3.5 | 87.2 ± 6.3 | 86.2 ± 7.0 | 85.9 ± 6.2 | 90.0 ± 2.6 | 90.0 ± 6.6 | 89.9 ± 5.2 | 88.9 ± 6.2 |
| | ALTER† | 88.9 ± 3.5 | 89.4 ± 4.6 | 88.0 ± 5.6 | 88.0 ± 5.0 | 90.9 ± 2.8 | 90.9 ± 4.6 | 90.1 ± 5.2 | 89.9 ± 4.4 |
| | BioBGT† | 81.5 ± 4.3 | 84.6 ± 3.6 | 82.1 ± 6.1 | 82.9 ± 3.7 | 83.2 ± 2.9 | 85.6 ± 7.3 | 82.9 ± 4.6 | 83.0 ± 7.1 |
| Hypergraph based model | HGNN | 83.7 ± 3.4 | 87.0 ± 2.6 | 80.0 ± 5.8 | 82.1 ± 4.1 | 85.1 ± 1.3 | 88.0 ± 1.9 | 84.5 ± 3.4 | 85.7 ± 2.5 |
| | HNHN | 87.6 ± 2.0 | 90.2 ± 2.0 | 85.1 ± 5.7 | 86.5 ± 4.3 | 86.9 ± 1.7 | 87.6 ± 2.7 | 86.5 ± 4.2 | 86.4 ± 3.0 |
| | UniGCNII | 89.4 ± 2.1 | 91.3 ± 1.6 | 87.1 ± 3.7 | 88.6 ± 2.5 | 88.3 ± 2.0 | 89.4 ± 2.6 | 86.3 ± 4.9 | 77.3 ± 4.1 |
| | dwHGCN† | 88.5 ± 3.0 | 90.6 ± 2.8 | 86.7 ± 4.3 | 88.1 ± 3.3 | 88.5 ± 2.1 | 91.3 ± 1.9 | 86.4 ± 4.8 | 88.2 ± 3.5 |
| | HyperGT | 82.5 ± 2.5 | 81.8 ± 9.8 | 70.5 ± 6.2 | 74.1 ± 7.4 | 81.2 ± 2.1 | 85.7 ± 7.1 | 69.8 ± 2.4 | 74.6 ± 3.6 |
| | HyBRiD† | 85.5 ± 5.4 | 87.3 ± 5.4 | 85.2 ± 6.9 | 85.5 ± 5.9 | 83.8 ± 4.4 | 83.5 ± 6.7 | 81.3 ± 6.1 | 82.1 ± 6.4 |
| | MuHL | **90.0 ± 2.1** | **93.4 ± 1.4** | **89.5 ± 6.4** | **89.2 ± 4.5** | **91.3 ± 1.9** | **93.2 ± 0.9** | **90.4 ± 4.1** | **90.8 ± 3.0** |

- **BrainNetTF** (Kan et al., 2022): We tuned the number of transformer heads $\{2, 4, 8\}$ and the number of clustering centers $\{2, 3, 4, 5, 10, 50, 100\}$ in the OCREAD readout.

- **ALTER** (Yu et al., 2024): We tuned the hop counts $k$ ranging from 2 to 16.

- **BioBGT** (Peng et al., 2025): We tuned the number of transformer heads $\{2, 4, 8\}$.

- **HyperDrop** (Jo et al., 2021): We set the hidden dimension of edges as 16, and randomly search for the edge drop ratio by increasing the drop ratio from 20% to 80%.

- **HyperGT** (Liu et al., 2024): We tuned the number of transformer heads $\{2, 4, 8\}$, and coefficient of balancing between classification and hyperedge structure losses $\{0.1, 0.3, 0.5, 0.7, 0.9\}$.

- **HyBRiD** (Qiu et al., 2024): We mainly tuned the key hyper-parameter such as the number of hyperedges $K \in \{4, 8, 12, 16, 20, 24, 28, 32\}$.

- **DHHNN** (Mei et al., 2025): We adjusted the hyperbolic curvature $c \in \{0.01, 0.1, 1\}$ and the hyperedge construction parameter $k \in \{8, 16, 32, 64, 128\}$.

For each baseline, the best configuration was selected based on validation performance. For hypergraph methods that are not inherently designed for brain networks (e.g., HGNN, HNHN, and HyperGT), the incidence matrix is not provided. Accordingly, the brain network hypergraph was constructed following the approach described in the HGNN paper and subsequently used for our experiments. In contrast, our method naturally learns the hypergraph structure in a data-driven manner, without relying on predefined construction rules.

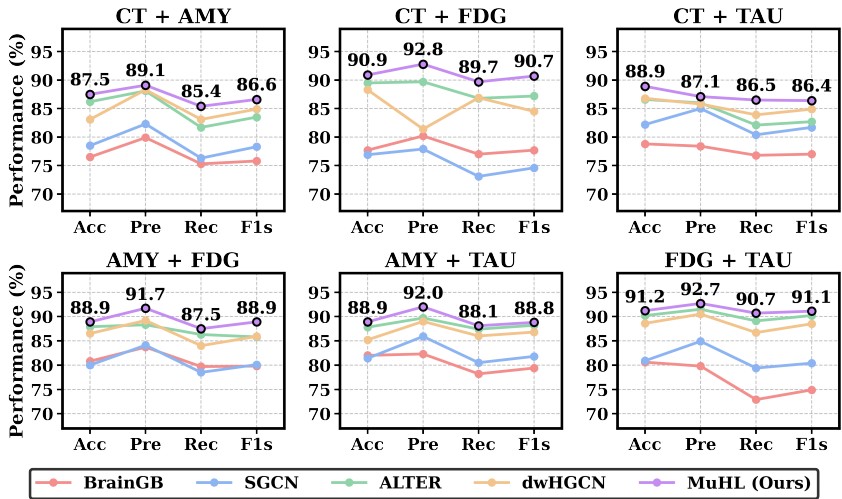

*Figure 6.* AD classification performance (CN/SMC/EMCI/LMCI/AD) on the ADNI dataset under various scenarios, especially for the combination of two modalities. The results compare our proposed MuHL with representative brain network analysis–based methods, highlighting the superior performance of MuHL.

*Table 8.* Left: Demographics of the ABIDE dataset. Right: Classification performance across hypergraph-based methods.

| Category | Control | Autism |
|---|---|---|
| # of subjects | 534 | 455 |
| Gender (M / F) | 441 / 93 | 399 / 56 |
| Age (Mean ± Std) | 16.89 ± 7.40 | 16.29 ± 7.58 |

| Method | ABIDE | | | |
|---|---|---|---|---|
| | Acc ↑ | Pre ↑ | Rec ↑ | F1s ↑ |
| HGNN | 56.5 ± 2.4 | 56.5 ± 2.1 | 56.4 ± 2.1 | 56.2 ± 2.3 |
| HNHN | 58.7 ± 0.8 | 58.5 ± 1.0 | 58.5 ± 1.1 | 58.4 ± 1.1 |
| UniGCNII | 60.0 ± 1.7 | 60.3 ± 2.2 | 59.0 ± 1.8 | 58.2 ± 2.6 |
| HyperDrop | 52.1 ± 4.1 | 49.0 ± 6.7 | 47.7 ± 8.3 | 46.9 ± 6.8 |
| dwHGCN | 55.2 ± 1.8 | 59.7 ± 9.4 | 53.1 ± 2.4 | 48.2 ± 7.0 |
| HyperGT | 56.5 ± 1.1 | 60.8 ± 5.5 | 54.8 ± 2.5 | 51.1 ± 7.2 |
| HyBRiD | 57.3 ± 2.6 | 60.1 ± 4.2 | 55.2 ± 3.1 | 57.5 ± 4.5 |
| DHHNN | 55.2 ± 3.5 | 58.8 ± 6.2 | 53.1 ± 4.6 | 54.6 ± 4.1 |
| **MuHL** | **63.7 ± 3.6** | **63.5 ± 3.7** | **63.3 ± 3.7** | **63.3 ± 3.7** |

## D. Additional Quantitative Results

**Effectiveness under Reduced-Modality and Dual-Modality Settings.** To further assess effectiveness in multi-modal analysis and evaluate the contribution of each imaging biomarker in the ADNI dataset, we conducted additional experiments by excluding one modality at a time from the complete combination of cortical thickness (CT), $\beta$-amyloid protein (AMY), Fluorodeoxyglucose (FDG), and tau protein (TAU). In Table 7, we compared performance between our method and some representative baselines. Across all four reduced-modality settings, MuHL consistently outperformed existing baselines in terms of classification performance. Despite the absence of one modality, our proposed method achieved the highest accuracy, precision, recall, and F1 score across every scenario, demonstrating its robustness under incomplete input conditions.

Additionally, we designed dual-modality fusion experiments on ADNI, covering six modality pairs, and the results are reported in Fig. 6. Across all combinations, MuHL consistently achieves the best performance on all evaluations when compared with representative brain-network baselines, i.e., BrainGB, SGCN, ALTER and dwHGCN. The gains are notable in Recall and F1-score, reflecting fewer false negatives while maintaining precision, which is particularly important for clinical screening. These improvements are attributed to multi-scale wavelet filtering that balances local–global cues and its data-driven hyperedge construction that models higher-order ROI interactions across complementary biomarkers. Importantly, the performance advantages are maintained even for biologically synergistic modality pairs (e.g., CT+FDG and AMY+TAU), highlighting that the benefits stem from the generality of the learned hypergraph structure rather than modality-specific effects.

The ability of MuHL to retain high classification performance under varying modality combinations highlights its practicality in real-world clinical settings. In many clinical environments, it is often difficult to obtain all imaging biomarkers for every patient due to constraints such as scan time, cost, or modality-specific availability. In such scenarios, relying on a model that performs well even with partial input becomes essential for broader deployment and equitable patient care.

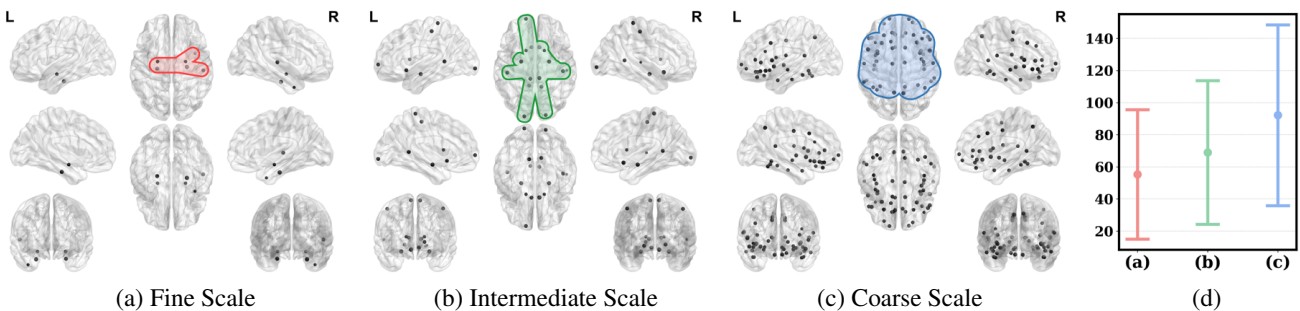

(a) Fine Scale      (b) Intermediate Scale      (c) Coarse Scale      (d)

*Figure 7.* **(a)-(c):** Visualization of a multi-resolution hypergraph on the ADNI dataset. At each scale, a representative hyperedge is shown, where the scale controls the granularity of high-order relations by adaptively adjusting how many nodes are included in the hyperedge. **(d):** The mean and standard deviation of the number of nodes per hyperedge at each scale, showing that node counts increase as the scale becomes coarser.

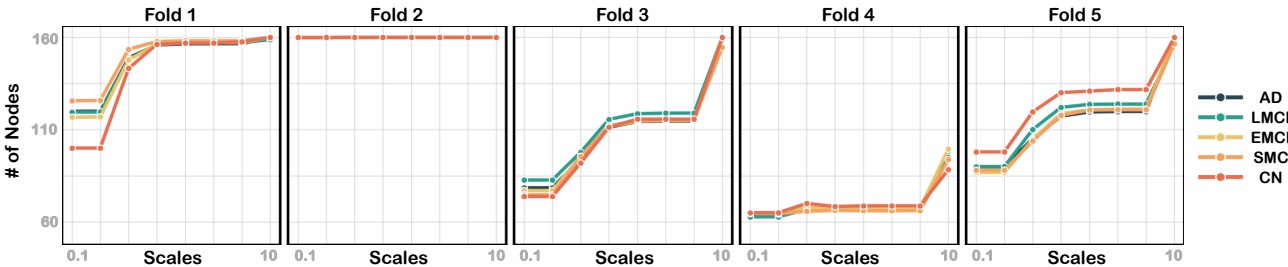

*Figure 8.* Class-wise salient hyperedge expansion across scales on the ADNI dataset. For each fold and diagnostic class, the most activated hyperedge is selected, and the number of nodes assigned to it is counted at each scale. The increasing node count from fine to coarse scales shows that MuHL progressively expands hyperedges from local ROI groups to broader high-order structures.

MuHL addresses this challenge by leveraging its multi-resolution design and data-driven hypergraph construction to extract meaningful patterns from whatever biomarker data are available. Rather than relying on rigid or predefined connectivity assumptions, the model adaptively learns higher-order relationships that are robust to missing modalities. This flexibility not only enhances its diagnostic utility but also aligns well with the variability and imperfection inherent in real-world medical data. Consequently, MuHL provides a generalizable framework for clinical decision support in neurodegenerative disease analysis, especially where comprehensive imaging protocols are infeasible.

**Robustness for Heterogeneous Disease.** To assess cross-domain robustness beyond neurodegenerative disorders such as AD and PD, we additionally evaluated our framework on an autism cohort. Autism Brain Imaging Data Exchange (ABIDE) (Heinsfeld et al., 2018) is used for our additional experiments, and the demographics is written in the left of Table 8. The ABIDE initiative aggregated functional brain imaging data collected from laboratories around the world to support the research on Autism Spectrum Disorder (ASD), which has stereotyped behaviors such as irritability, hyperactivity, depression, and anxiety. As in the right of Table 8, the classification results across hypergraph-based baselines. Our model achieves the best performance on every metric, and UniGCNII records the second-best accuracy and recall. These results indicate that MuHL generalizes across heterogeneous brain disorders, spanning neurodegenerative (i.e., AD/PD) and neurodevelopmental (ASD) conditions, and provides meaningful cross-disease representations.

## E. Additional Discussions on the Learned Hypergraph

**Multi-Scale Hyperedge Generation.** To clarify the effect of multi-scale optimization on hypergraph structure, as shown in Fig. 7, a representative hyperedge is constructed at different scales on the ADNI dataset. After training, each scale-wise hyperedge represents how the scope of connectivity and node participation in high-order structures change across scales. To visualize of multi-resolution hyperedge construction in Fig. 7, since each node–hyperedge pair is associated with a continuous assignment score, we apply a uniform threshold across all scales to discretize these scores into binary connections, which allows us to characterize connectivity. As the scale increases from (a) to (c), the number of nodes assigned to the hyperedge grows, indicating that higher scales capture broader and more global interactions.

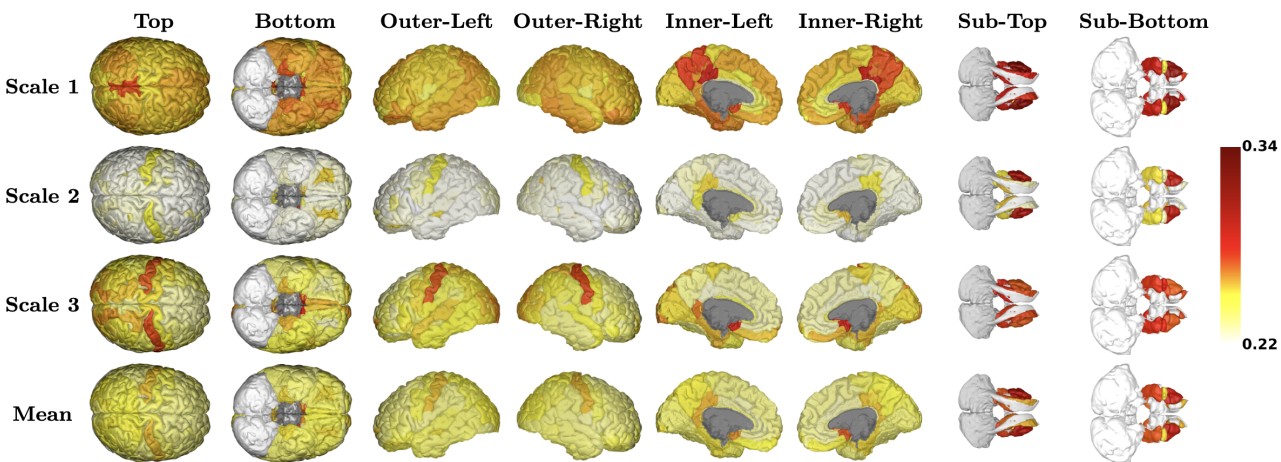

*Figure 9.* Visualization of cumulative hyperedge activations on cortical/subcortical regions from ADNI dataset using all biomarkers. The top three rows represent hyperedge activations at each scale, while the bottom row shows their mean.

Also, (d) summarizes hyperedge size statistics for each scale by aggregating across all hyperedges and the five folds. We report the mean and standard deviation of the number of nodes per hyperedge, revealing a clear monotonic increase from the fine to the coarse scale. This upward shift indicates that larger scales capture broader group-wise interactions, consistent with the qualitative examples in from (a) to (c). Therefore, these results highlight the ability of MuHL to flexibly capture group-wise interactions across scales, enabling the construction of expressive and biologically meaningful high-order structures from brain networks.

**Class-wise Expansion of Salient Hyperedges across Scales.** To further examine whether the learned hyperedges consistently expand across scales for each disease stage, we analyze the most salient hyperedge in a class-wise manner. For each fold and diagnostic class, we select the hyperedge with the largest aggregated node activation and measure the number of nodes assigned to that hyperedge as the scale increases. As shown in Fig. 8, the number of included nodes generally increases from fine to coarse scales across folds and classes. This indicates that fine-scale hyperedges capture localized ROI groups, whereas coarse-scale hyperedges progressively cover broader brain regions. The trend is consistently observed for CN, SMC, EMCI, LMCI, and AD, suggesting that MuHL learns scale-dependent high-order structures in a robust manner rather than producing class- or fold-specific artifacts. These results further support the dilation behavior of learned hyperedges described in Proposition 2.

**Key ROIs via Hyperedge Activations.** Understanding the importance of ROIs in brain networks is essential for AD progression analysis, as these regions play a crucial role in characterizing disease-related alterations. The aggregated hyperedge activations (i.e., degree) for each ROI serve as an indicator of its network significance. As shown in Fig. 9, ROIs with high cumulative weights act as network hubs, facilitating information flow and exhibiting greater sensitivity to pathological changes in AD. While the main paper highlights the top 10 ROIs, Fig. 9 presents the cumulative hyperedge activations of all ROIs across multiple perspectives, providing a comprehensive view of their importance. Identifying these key ROIs highlights disease-relevant regions most affected by neurodegeneration, enhancing both model interpretability and clinical relevance.

**Additional Case Studies for Key Hyperedges.** To further examine the biological relevance of the learned hyperedges, we analyzed additional highly activated hyperedges obtained from MuHL trained on the ADNI dataset. The hyperedges with the strongest aggregated activation were selected for in-depth inspection. As shown in Fig. 10, the two representative hyperedges exhibit coherent spatial patterns: several ROIs repeatedly emerge across different hyperedges (e.g., hippocampus, amygdala, postcentral gyrus, and subcallosal regions), and many appear as symmetric left–right pairs, reflecting well-known bilateral organization in AD-related atrophy and functional disruption. These patterns also form anatomically plausible groupings, such as limbic–striatal circuits and cingulo-parietal pathways, which have been consistently implicated in episodic memory decline, emotional processing deficits, and large-scale network disintegration in AD.

In addition, the configuration of these hyperedges captures both localized clusters of ROIs and long-range inter-regional co-activations, indicating that MuHL successfully identifies higher-order group interactions beyond simple pairwise connectivity. Taken together, these case studies demonstrate that the most salient hyperedges discovered by MuHL are not only statistically

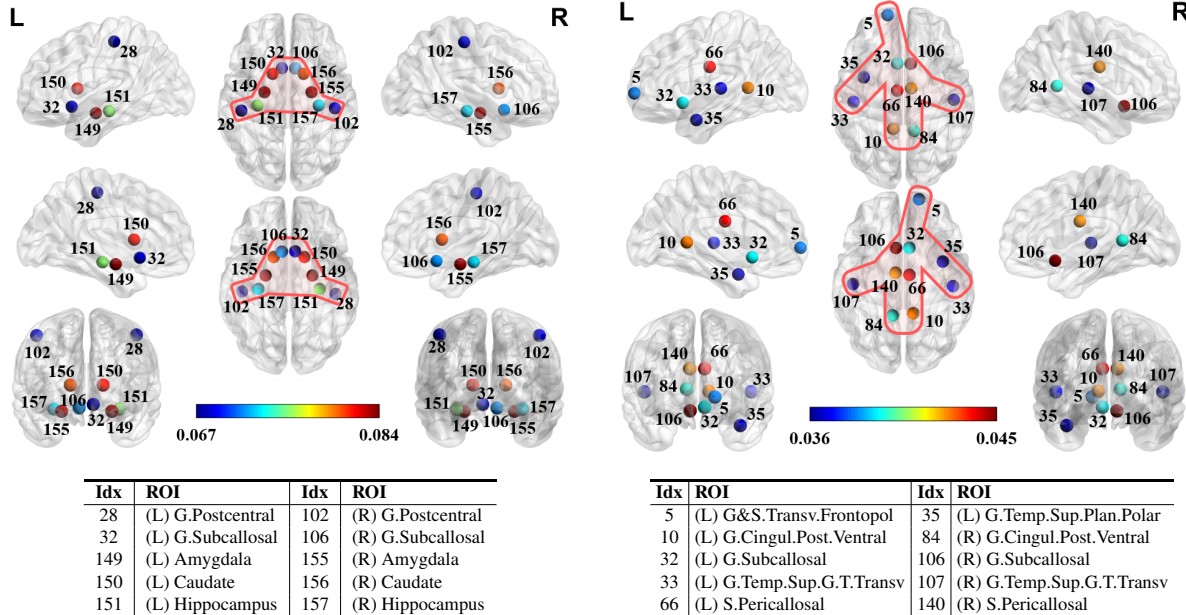

*Figure 10.* Top: Visualization of the top-10 ROIs associated with additional important hyperedges in the ADNI studies, based on the sum of node activations for each hyperedge. Node color indicates activation value (i.e., weight) per node within the hyperedge. Bottom: Corresponding ROI labels.

*Table 9.* Statistical consistency of ROI importance across independent runs on the ADNI and PPMI datasets. The mean values of Kendall's $\tau$, p-values, and Jaccard similarity for varying numbers of runs are reported.

| Dataset | | ADNI | | | | PPMI | | | |
|---|---|---|---|---|---|---|---|---|---|
| Independent Runs | | 5 | 10 | 20 | 30 | 5 | 10 | 20 | 30 |
| All ROIs | Kendall's $\tau$ | 0.798 | 0.796 | 0.767 | 0.750 | 0.721 | 0.686 | 0.654 | 0.645 |
| | P-value | 1.13e-18 | 5.42e-16 | 3.79e-11 | 2.99e-11 | 1.09e-14 | 1.15e-10 | 1.41e-06 | 1.07e-05 |
| | Kendall's $W$ | 0.871 | 0.866 | 0.838 | 0.821 | 0.834 | 0.762 | 0.734 | 0.713 |
| Top-20 ROIs | Jaccard Similarity | 0.893 | 0.881 | 0.835 | 0.817 | 0.824 | 0.770 | 0.746 | 0.735 |

stable across runs but also biologically meaningful, aligning with established AD-related neural circuits reported in prior neuropathological and neuroimaging studies.

**Statistical Validation of ROI-level Stability.** To evaluate whether the ROIs highlighted by MuHL reflect reproducible disease-related patterns rather than artifacts of random initializations, we performed a stability analysis across multiple independent runs with different random seeds, summarized in Table 9. For each run, the learned incidence matrix was aggregated into ROI-level importance scores, and concordance across runs was evaluated using three complementary metrics. First, *Kendall's $\tau$ coefficients* (Sen, 1968) remained consistently high ($> 0.6$) across ADNI and PPMI, with extremely small p-values ($< 10^{-5}$). This indicates that the ordering of ROI importance is highly unlikely to arise by chance. Second, *Kendall's $W$* (Legendre, 2005), which measures agreement across all runs jointly, exceeded 0.80 on ADNI and 0.70 on PPMI, reflecting stable consistency in overall importance rankings. Finally, *Jaccard similarity* (Niwattanakul et al., 2013) of top-20 ROIs remained above 0.80 for ADNI and above 0.73 for PPMI, confirming that the most discriminative ROIs are robustly preserved across runs. These results show that the ROI patterns captured by MuHL are statistically stable and not attributable to random fluctuations, thereby reinforcing the interpretability and biological reliability of the learned hypergraph representations.

# F. Prediction ability for AD clinical Symptoms

We further employed the Lasso linear regression (Tibshirani, 1996) to predict AD-related clinical symptoms including CDR-SOB, ADAS11, ADAS13, and MMSE (Mohs & Cohen, 1988; O'Bryant et al., 2008; Delor et al., 2013; Balsis et al., 2015; Lowe et al., 2015; Kueper et al., 2018). This analysis aims to elucidate the relationship between structural/functional brain alterations and clinical scores, offering insights into the potential and limitations of imaging-based predictive models

*Table 10.* Regression comparison between brain network-based methods and MuHL to predict AD symptoms. The evaluation metrics between the predicted and true scores are CDR-SOB, ADAS11, ADAS13 and MMSE on the ADNI dataset.

| Clinical Measures | Methods | Metrics | | | |
|---|---|---|---|---|---|
| | | P value ↓ | Correlation ↑ | RMSE ↓ | R Squared ↑ |
| CDR-SOB | BrainGB | 4.66e-09 | 0.4857 | 0.7799 | 0.6254 |
| | BrainNetTF | 1.22e-09 | 0.5014 | 0.7900 | 0.6717 |
| | SGCN | 2.38e-08 | 0.4655 | 0.8000 | 0.7131 |
| | ALTER | 7.86e-10 | 0.5064 | 0.7947 | 0.6299 |
| | MuHL(Ours) | 2.76e-09 | 0.4920 | 0.7625 | 0.7204 |
| ADAS11 | BrainGB | 1.38e-17 | 0.6597 | 2.8198 | 0.7517 |
| | BrainNetTF | 1.75e-20 | 0.7004 | 2.7274 | 0.4937 |
| | SGCN | 2.60e-13 | 0.5853 | 3.0907 | 0.8470 |
| | ALTER | 7.92e-16 | 0.6315 | 2.9718 | 0.4437 |
| | MuHL(Ours) | 1.43e-21 | 0.7701 | 2.5205 | 0.8694 |
| ADAS13 | BrainGB | 4.99e-19 | 0.6808 | 4.5031 | 0.5418 |
| | BrainNetTF | 7.29e-20 | 0.6923 | 4.3246 | 0.5197 |
| | SGCN | 3.82e-15 | 0.6196 | 4.7065 | 0.5936 |
| | ALTER | 7.53e-18 | 0.6637 | 4.5317 | 0.5219 |
| | MuHL(Ours) | 5.61e-20 | 0.6340 | 4.2403 | 0.6219 |
| MMSE | BrainGB | 4.00e-04 | 0.3059 | 1.5841 | 0.5296 |
| | BrainNetTF | 6.70e-07 | 0.4196 | 1.5080 | 0.5572 |
| | SGCN | 1.48e-03 | 0.3266 | 1.5809 | 0.6305 |
| | ALTER | 2.85e-05 | 0.3582 | 1.5435 | 0.3272 |
| | MuHL(Ours) | 8.27e-08 | 0.4492 | 1.4953 | 0.6265 |

in clinical contexts. To predict each of these clinical measures, we leveraged the cognitive decline representations of the last layer derived from trained MuHL, which is compared with other baselines for brain analysis. These representations were normalized to a unit interval and subsequently used to train the regression approach via the 5-fold cross validation.

In Table 10, we reported the performance comparison between brain network-based methods and MuHL. We used various metrics such as Pearson correlation coefficient, p-value, root mean squared error (RMSE) and R-squared score to evaluate the effectiveness of our trained model in predicting clinical scores compared to other brain-based models. MuHL consistently showed improved performance over other baselines across all metrics, particularly in the R-squared score; exceeding previous best baseline results by 0.73%p (on CDR-SOB), 2.24%p (on ADAS11) and 2.83%p (on ADAS13). These high R-squared measures indicate that MuHL effectively captures the variance in the data, enhancing its reliability in predicting various AD-related clinical features accurately. Thus, the neurodegenerative representations of MuHL demonstrated strong predictive performance across all clinical measures, highlighting their underlying associations with AD symptoms.

# G. Additional Ablation Studies

## G.1. Stability of Learned Hyperedges and Initialization Robustness

**Behavior of Learned Hyperedges.** To examine how the learned hyperedges behave under different folds and initialization schemes such as Xavier (Glorot & Bengio, 2010) and Kaiming He (He et al., 2015), we first visualize the optimized incidence matrices in Fig. 11. For a fixed dataset and training configuration, the hyperedges converge toward a similar underlying structure during optimization, leading to incidence matrices with comparable activation patterns across initialization schemes. Reflecting the inherent biological topology of the brain, the multi-scale filtered node features $X_s$ present dominant and consistent patterns, which enable the learning process to rapidly adjust the randomly initialized $\Phi$ toward this convergence. Across folds, the model is trained on different subsets of subjects, and the hyperedges do not have a fixed ordering, so directly matching individual hyperedge columns across folds is not meaningful. Nevertheless, the overall sparsity and the distribution of active regions remain similar, indicating that the structural behavior of the learned hypergraph is largely consistent across all folds.

Fig. 11 further shows that these patterns are remarkably stable across both cross-validation folds and initialization schemes, revealing a similar arrangement of dominant and inactive regions regardless of how training begins or how the dataset is partitioned. This reproducibility is supported by the multi-scale wavelet filtering, which produces smoothed node representations whose dominant relational structure is shaped by the underlying graph geometry. In addition, the subsequent operations, including the use of ReLU, row-wise SoftMax normalization, and Top-K sparsification, suppress small perturbations arising

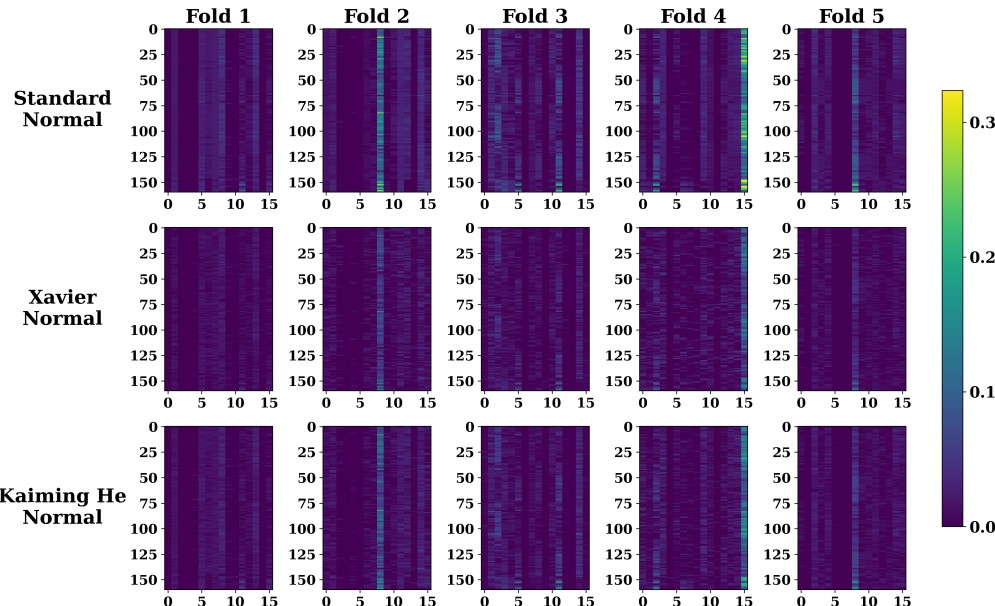

*Figure 11.* Incidence matrices of learned hyperedges for three initialization schemes of Φ across five folds. The similar activation patterns across initialization schemes indicate that the learned hypergraph structure is robust to the choice of initialization.

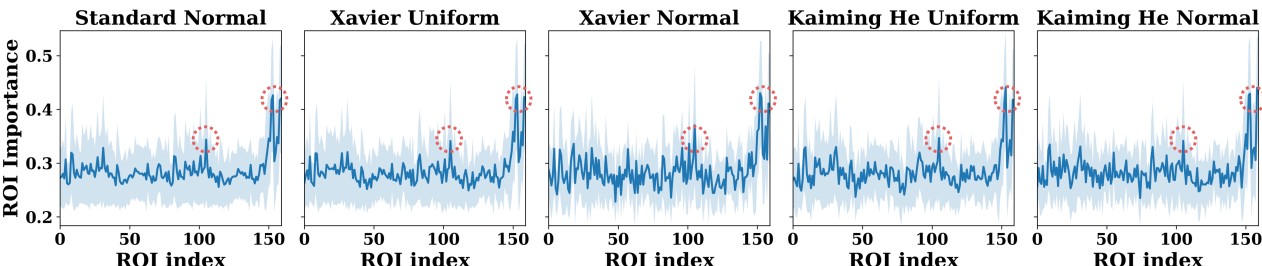

*Figure 12.* Aggregated ROI importance over five folds for five weight initialization schemes of the hyperedge projection Φ. The main peaks and overall patterns remain consistent, indicating that the learned hyperedges and resulting ROI level importance are robust to both folds and initialization.

*Table 11.* Classification performance of MuHL under five initialization schemes for the hyperedge projection matrix Φ on the ADNI and PPMI datasets.

| Initialization | ADNI | | | | PPMI | | | |
|---|---|---|---|---|---|---|---|---|
| | Acc ↑ | Pre ↑ | Rec ↑ | F1s ↑ | Acc ↑ | Pre ↑ | Rec ↑ | F1s ↑ |
| Xavier Uniform | $92.0 \pm 2.4$ | $93.8 \pm 3.9$ | $92.7 \pm 2.3$ | $92.2 \pm 3.0$ | $73.7 \pm 3.1$ | $63.1 \pm 6.3$ | $58.3 \pm 5.5$ | $60.6 \pm 6.0$ |
| Xavier Normal | $92.5 \pm 2.1$ | $94.7 \pm 2.0$ | $93.9 \pm 2.5$ | $94.2 \pm 2.5$ | $74.3 \pm 4.5$ | $64.1 \pm 6.8$ | $59.4 \pm 5.4$ | $61.6 \pm 5.9$ |
| Kaiming He Uniform | $92.2 \pm 3.3$ | $93.2 \pm 5.4$ | $92.5 \pm 5.9$ | $92.8 \pm 5.5$ | $74.2 \pm 4.2$ | $63.1 \pm 5.6$ | $59.3 \pm 5.2$ | $61.1 \pm 5.4$ |
| Kaiming He Normal | $92.1 \pm 3.3$ | $93.5 \pm 4.6$ | $92.5 \pm 5.9$ | $92.8 \pm 5.5$ | $75.2 \pm 5.7$ | $64.3 \pm 5.5$ | $60.2 \pm 5.2$ | $62.1 \pm 5.3$ |
| Standard Normal | $\mathbf{93.2 \pm 2.4}$ | $\mathbf{95.4 \pm 1.3}$ | $\mathbf{94.2 \pm 1.6}$ | $\mathbf{94.7 \pm 1.5}$ | $\mathbf{76.8 \pm 3.7}$ | $\mathbf{66.6 \pm 7.6}$ | $\mathbf{60.7 \pm 6.0}$ | $\mathbf{62.4 \pm 6.6}$ |

from initialization and emphasize persistent node–hyperedge affinities. Thus, the model consistently recovers similar high-order interaction patterns, reflecting the intrinsic stability and robustness of the hyperedge learning process against varying initial conditions.

**Stability of ROI-wise Importance.** For interpretability, the primary criterion is the stability of ROI-level importance, given that the ROIs are fixed and directly comparable across runs. To examine this stability, Fig. 12 summarizes ROI importance by aggregating hyperedge activations within each ROI computed by summing each row of the incidence matrices and averaging over folds for each intialization scheme. The results in Fig. 12 exhibits highly consistent trends across all five initialization schemes, with the same ROIs repeatedly forming the dominant peaks. This consistency demonstrates that, irrespective of the initial values of the hyperedge projection matrix, MuHL reliably identifies a common set of disease-related ROIs, indicating

*Table 12.* Classification performance of MuHL with different combinations of low-pass and band-pass spectral filters on the ADNI and PPMI datasets.

| Number of Filters | | ADNI | | | | PPMI | | | |
| --- | --- | --- | --- | --- | --- | --- | --- | --- | --- |
| Low pass | Band pass | Acc ↑ | Pre ↑ | Rec ↑ | F1s ↑ | Acc ↑ | Pre ↑ | Rec ↑ | F1s ↑ |
| 3 | 0 | $92.2 \pm 2.4$ | $94.4 \pm 1.8$ | $91.4 \pm 5.3$ | $92.4 \pm 4.0$ | $69.6 \pm 1.8$ | $62.0 \pm 6.1$ | $50.0 \pm 4.7$ | $55.3 \pm 5.1$ |
| 0 | 3 | $91.5 \pm 1.9$ | $94.6 \pm 0.8$ | $91.3 \pm 3.1$ | $92.6 \pm 1.8$ | $72.9 \pm 3.3$ | $62.3 \pm 8.2$ | $56.8 \pm 4.5$ | $58.4 \pm 5.8$ |
| 1 | 2 | $\mathbf{93.2 \pm 2.4}$ | $\mathbf{95.4 \pm 1.3}$ | $\mathbf{94.2 \pm 1.6}$ | $\mathbf{94.7 \pm 1.5}$ | $\mathbf{76.8 \pm 3.7}$ | $\mathbf{66.6 \pm 7.6}$ | $\mathbf{60.7 \pm 6.0}$ | $\mathbf{62.4 \pm 6.6}$ |

*Table 13.* Comparison of MuHL variants with static or dynamic hyperedge structure and hyperedge weights, including a dwHGCN style weight-only baseline, evaluated on the ADNI dataset. Values in parentheses indicate performance changes relative to dwHGCN.

| Method | Hyperedge structure | Hyperedge weights | ADNI | | | |
| --- | --- | --- | --- | --- | --- | --- |
| | | | Acc ↑ | Pre ↑ | Rec ↑ | F1s ↑ |
| dwHGCN | Static | Dynamic | $90.2 \pm 1.3$ | $87.2 \pm 7.1$ | $86.3 \pm 6.8$ | $86.2 \pm 6.4$ |
| MuHL | Static | Static | $88.0 \pm 2.6$ $(-2.2)$ | $88.7 \pm 3.3$ $(+1.5)$ | $87.9 \pm 4.3$ $(+1.6)$ | $87.9 \pm 3.6$ $(+1.7)$ |
| MuHL | Static | Dynamic | $89.1 \pm 2.3$ $(-1.1)$ | $88.1 \pm 3.8$ $(+0.9)$ | $87.4 \pm 4.0$ $(+0.9)$ | $87.3 \pm 3.7$ $(+0.9)$ |
| MuHL | Dynamic | Static | $\mathbf{93.2 \pm 2.4}$ $(+3.0)$ | $\mathbf{95.4 \pm 1.3}$ $(+8.2)$ | $\mathbf{94.2 \pm 1.6}$ $(+7.9)$ | $\mathbf{94.7 \pm 1.5}$ $(+8.5)$ |

that the model provides a robust and meaningful representation of important brain regions.

**Effect of Initialization Schemes on Performance.** Finally, to evaluate how initialization affects model performance, Table 11 reports results across five standard initialization schemes. In our framework, $\Phi$ is treated as a standard learnable parameter initialized with a zero-mean unit-variance Gaussian distribution, without imposing additional structural constraints such as sparsity, non-negativity, or anatomical priors. Its outputs are then passed through ReLU and a row-wise Softmax to construct the incidence matrix, which normalizes scale differences arising from initialization and enables the hypergraph structure to be learned end-to-end from the data. Across all initialization schemes, the overall performance remains effectively unchanged, and the gains over baseline methods are consistently preserved. These results indicate that the model is largely insensitive to the specific choice of initialization, and the performance improvements primarily stem from the proposed multi-scale hypergraph architecture rather than from parameter initialization.

## G.2. Spectral Filter Composition and Multi-Resolution Design Choice

To justify the use of one low-pass and multiple band-pass spectral filters in our multi-scale construction, we compare alternative filter configurations in Table 12. Conventional graph wavelet theory (Hammond et al., 2011) motivates this design by emphasizing that global structure should be captured by a single smoothing (low-pass) component, while finer structural variations require multiple band-pass components operating at different resolutions. To further verify that this hybrid configuration is necessary, we additionally tested two alternative settings: (1) using only low-pass filters across all scales, and (2) using only band-pass filters. Using only low-pass filters substantially reduces performance due to excessive smoothing across scales, whereas using only band-pass filters removes the global structural anchor needed for stable hyperedge construction. In contrast, the proposed combination of one low-pass and multiple band-pass filters produces the best results on both ADNI and PPMI datasets, demonstrating that multi-resolution analysis with a balanced mixture of global and localized spectral responses is essential for effective representation learning in MuHL.

## G.3. Effect of Dynamic Hypergraph Structure Learning

To examine the contribution of dynamic hyperedge structure learning, we evaluated MuHL variants that selectively disable either hyperedge structure updates or hyperedge weights updates, and compared them with a dwHGCN (Wang et al., 2023). As shown in Table 13, learning hyperedge structure dynamically yields the largest performance improvement across all metrics. When both the hyperedge structure and parameters are kept static, performance drops notably, indicating that fixed hyperedges cannot capture subject-specific higher-order interactions. Allowing only the hyperedge parameters to update provides limited gains, suggesting that weight refinement alone is insufficient without adjusting the underlying node to hyperedge assignments. In contrast, enabling dynamic structure learning while fixing the hyperedge weights still produces a substantial boost, demonstrating that adaptively reorganizing ROI groupings is the primary driver of discriminative power. These results confirm that the core advantage of MuHL lies in learning continuous and multi-scale hyperedge structures,

which cannot be achieved by weight updates alone.

## H. Model Complexity

Table 14 presents a comparison of efficiency across representative hypergraph-based methods. In our design, the hyperedges for high-order modeling increase parameter counts, and FLOPs are mainly driven by spectral decomposition in multi-scale filtering. Despite this, MuHL achieves a runtime of 9.98 ms, which is substantially faster than HyperGT (97.06 ms) and still comparable to lighter baselines such as HGNN and dwHGCN. Although this multi-scale hyperedge modeling raises memory usage, all methods, including MuHL, run smoothly on a single GPU, ensuring reproducibility. These findings demonstrate that the computational overhead required to capture high-order relationships does not lead to prohibitive latency or practical limitations. Instead, MuHL achieves a balanced trade-off between structural modeling cost and runtime efficiency, while delivering state-of-the-art performance and reinforcing its value for real-world neurodegenerative disease analysis.

*Table 14.* Comparison of efficiency between MuHL and representative hypergraph-based methods. Efficiency is evaluated in terms of the number of parameters, FLOPs, and average inference runtime across 5-folds.

| Efficiency | Methods | | | | |
|---|---|---|---|---|---|
| | HGNN | dwHGCN | HyperGT | HyBRiD | MuHL |
| Params (M) | 3.28 | 3.28 | 3.29 | 4.31 | 5.54 |
| FLOPs (G) | 0.43 | 0.43 | 0.52 | 0.69 | 0.88 |
| Runtime (ms) | 1.84 | 1.72 | 97.06 | 22.13 | 9.98 |

Formally, the overall complexity consists of three main components in MuHL: 1) graph wavelet transform with $O(KN)$ using Chebyshev approximation, 2) hypergraph convolution with $O(NMd)$, and 3) multi-head self-attention with $O(JN^2d)$, where $N$ denotes the number of nodes, $M$ is the number of hyperedges, $d$ represents the feature dimension, $J$ is the number of scales, and $K$ indicates the wavelet order used in the Chebyshev approximation. Importantly, each component can be efficiently approximated or applied sparsely, thereby maintaining efficiency comparable to standard GNNs. In our current experiments, MuHL is implemented without such approximations yet already achieves practical efficiency. Thus, future incorporation of approximation techniques is expected to further enhance its computational efficiency.

## I. Discussions on Our Work

**Limitations.** Although our method adaptively learns meaningful multi-resolution representations, the choice of the number of scales $J$ remains a dataset-dependent hyperparameter that may require empirical tuning for optimal performance. Additionally, the integration of multi-scale filtering and hypergraph learning leads to increased memory usage and computation compared to conventional GNNs. However, since wavelet filtering and hypergraph convolutions across scales are performed in parallel, the overall runtime remains comparable. Furthermore, during inference, the model maintains efficiency as scale-wise operations are precomputed and reused. These considerations indicate that while MuHL incurs extra computational overhead, it maintains a balanced trade-off between representational capacity and practical efficiency, ensuring feasibility for real-world scenarios.

**High-Order Modeling in Brain Networks.** Brain dysfunction in diverse neurodegenerative disorders such as AD and PD arises from coordinated alterations across multiple ROIs, rather than from isolated pairwise connections. In the brain, complex functions often rely on pathways that span multiple intermediate synapses, on hub regions that gather and redistribute information, and on synchrony across spatially distant areas. These are all collective phenomena that cannot be fully explained by isolated pairwise connections. Pairwise GNNs approximate such effects only indirectly and often at the cost of oversmoothing. In contrast, hyperedges model multi-ROI dependencies explicitly, and the proposed wavelet coupling across multiple scales captures how these dependencies evolve with scale. In this domain, the biological meaning of a hyperedge as a set of jointly altered ROIs and its scale is well defined and can be validated against known circuits and clinical outcomes. This makes brain networks a particularly compelling and necessary application area for high order structure learning, beyond what a generic graph formulation can provide.

**LLMs Usage.** We employed LLMs to aid in polishing the writing of this manuscript. No part of the research design, experiments, or analysis was generated by the LLM.

