# OpenReview forum: "Learning Multi-Scale Hypergraph for High-Order Brain Connectivity Analysis"
_ICML.cc/2026/Conference — ICML 2026 regular_

### Official Review · Reviewer_Uken · 2026-03-03

**Soundness:** 3
**Presentation:** 4
**Significance:** 3
**Originality:** 3
**Overall Recommendation:** 5
**Confidence:** 3

**Summary:**

This paper introduces MuHL, an adaptive Multi-scale Hyperedge Learning framework for high-order brain connectivity analysis using hypergraphs. The approach combines graph wavelet transform-based multi-resolution feature extraction and learnable hyperedge construction to explicitly model group-wise interactions between brain regions of interest (ROIs). Theoretical analysis establishes the connection between wavelet scale and hyperedge dilation. The authors validate MuHL’s effectiveness in classifying neurodegenerative diseases (Alzheimer’s and Parkinson’s) on multiple public brain network benchmarks, demonstrating significant improvements over state-of-the-art graph- and hypergraph-based methods. The paper also emphasizes interpretability through the identification of key ROIs and their disease-relevant interactions.

**Compliance With Llm Reviewing Policy:**

Affirmed.

**Key Questions For Authors:**

1. Can authors provide deeper biological or neuroscientific validation for the discovered hyperedges (beyond activation/importance)? How closely do MuHL’s groupings correspond to known disease modules or clinical findings, especially for Alzheimer’s and Parkinson’s?

1. When performing Zero-Shot Learning, is the model strictly limited to the same disease, same modality of data, and same number of nodes? Otherwise, would it lead to a decrease in accuracy (e.g., a model trained on AD data being applied to PD classification) or even become completely unusable (e.g., a model trained on the AAL atlas being applied to the BN atlas)?

1. Can the authors make the related code open source to enhance reproducibility?

**Limitations:**

yes

**Strengths And Weaknesses:**

**Strengths**
1. The paper presents a principled and well-integrated framework that unifies multi-resolution spectral feature extraction and adaptive hypergraph learning.
1. Theoretical results (Propositions 1 and 2) clarify how the multi-scale construction underpins dynamic hyperedge dilation and connectivity, tightly connecting intuition and mathematical rigor.
1. Extensive experiments on large, multi-modal neuroimaging datasets (ADNI and PPMI) show consistent, significant performance gains in disease-stage classification and robustness across modalities and settings.
1. Ablation, sensitivity, and robustness studies are thorough; interpretations are backed by well-constructed qualitative visualizations (e.g., Figures 1, 3, 4, 7) and tables (e.g., Tables 1, 2, 4, 12, 13).
1. Interpretability is directly addressed, with ROI importance analysis and reproducibility of learned structures explored in depth, which is meaningful for medical reaserch and applications.
1. The supplementary material is comprehensive, including additional datasets, implementation, and hyperparameter details.

**Weaknesses**
1.  The "multi-scale transformer" stage reuses standard transformer components. It would be helpful to clarify what is uniquely contributed (if anything) compared to conventional multi-head attention GNNs.
1. The interpretability discussion focuses on cumulative hyperedge activations and ROI rankings (Figures 3, 8, 9). However, the current biological interpretation, while plausible, is somewhat shallow. Connections to network neuroscience or external biological validation (e.g., comparison to clinical domain knowledge beyond standard atlases) could deepen the impact.
1. Table 14 shows higher parameterization and FLOPs for MuHL relative to some baselines; the authors mention this but do not deeply engage with potential scalability bottlenecks in the context of even larger datasets or finer-meshed network data.

---

> ### Author Rebuttal · Authors · 2026-03-27
>
> We sincerely thank the reviewer for the careful and constructive comments and for the positive score and encouraging assessment of our work. We will clear out all the remaining questions in the following.
>
> ---
>
> **[W1] Role of multi-scale transformer**
>
> [A] The Transformer block itself is standard, and our claim of novelty does **not lie in the self-attention mechanism alone, but in its functional incorporation within MuHL.** Conventional multi-head attention GNNs usually apply multiple attention projections over a fixed pairwise graph structure. In contrast, MuHL applies the Transformer to a sequence of embeddings that already encode scale-specific hypergraph structure learned from wavelet filtered features. Consequently, the Transformer in MuHL incorporates higher-order relational information across scales, rather than simply reweighting neighborhoods. Our ablation study further supports that the performance gain stems from this coupled design rather than from attention alone. Specifically, removing the multi-scale transformer degrades performance, while removing hypergraph structure learning causes the largest drop. Thus, this suggests that the main novelty lies in the coupled design rather than in the reuse of attention alone.
>
> ---
>
> **[W2/Q1] Depth of biological interpretation**
>
> [A] We appreciate this point and agree that stronger neuroscience-grounded validation would further improve the paper. Beyond ROI ranking or activation magnitude in the manuscript, we already observe that the learned hyperedges recover clinically meaningful AD and PD-related circuits rather than arbitrary ROI sets.
> For example, in AD from Fig. 4, the identified hyperedges effectively link the posterior cingulate and precuneus with the hippocampus, a finding that aligns with the established functional coupling between the posterior default mode network and the medial temporal memory system [1,2]. In PD, the most salient pattern is the cluster formed by the thalamus, insula, and amygdala, which corresponds to the basal-ganglia–thalamocortical circuit widely implicated in both motor and non-motor symptoms of PD [3,4].
> These patterns indicate that MuHL captures disease-related higher-order groupings that are consistent with prior clinical and neuroscientific findings. In the revision, we will make these links to known disease modules and network neuroscience more explicit so that the biological interpretation goes beyond plausibility and is better grounded in established domain knowledge.
>
> [1] Miao, Xiaoyan, et al. "Altered connectivity pattern of hubs in default-mode network with Alzheimer's disease: an Granger causality modeling approach." PloS one 2011.
>
> [2] Menon, Vinod. "20 years of the default mode network: A review and synthesis." Neuron 2023.
>
> [3] Galvan, Adriana, et al. "Alterations in neuronal activity in basal ganglia-thalamocortical circuits in the parkinsonian state." Frontiers in neuroanatomy 2015.
>
> [4] Banwinkler, Magdalena, et al. "Imaging the limbic system in Parkinson’s disease—A review of limbic pathology and clinical symptoms." Brain sciences 2022.
>
> ---
>
> **[W3] Computational scalability**
>
> [A] The additional computation in MuHL does not represent a practical bottleneck in the current setting. Although Table 14 shows higher parameter count and FLOPs than lightweight hypergraph baselines, MuHL still maintains practical runtime at current brain graph sizes and is more efficient in inference than some stronger transformer-based competitors (e.g., HyperGT). In addition, its main operations are compatible with sparsification, approximation, and parallelization, which support scalability toward finer-grained networks. MuHL therefore trades moderate additional computation for richer higher-order modeling while remaining practical in the current setting and admitting a clear path toward further scaling.
>
> ---
>
> **[Q2] Scope of zero-shot transfer**
>
> [A] In the current paper, zero-shot transfer is considered only under compatible graph representations, meaning that the source and target datasets share the same disease context, data modality, and atlas-based node definition. This is not a restriction specific to MuHL, but a common assumption in most brain network-based models built on fixed graph representations. When the disease label space changes completely, or when the atlas changes so that the number or identity of nodes differs, direct transfer generally becomes invalid without additional alignment or adaptation. Our zero-shot claim is therefore focused on realistic cross-cohort and cross-acquisition transfer under consistent graph construction.
>
> ---
>
> **[Q3] Code availability and reproducibility**
>
> [A] We plan to release the code and the training configuration used in the paper upon publication, including the main hyperparameter settings and evaluation protocol, so that the results can be more easily verified and extended by the community.

---

> > ### Author Rebuttal · Reviewer_Uken · 2026-04-04
> >
> > Thank you for your response. I will keep my score unchanged

---

### Official Review · Reviewer_dcFp · 2026-03-05

**Soundness:** 3
**Presentation:** 4
**Significance:** 4
**Originality:** 4
**Overall Recommendation:** 5
**Confidence:** 2

**Summary:**

This paper proposes MuHL, a framework that combines multi-resolution spectral graph wavelet filtering with adaptive hypergraph construction to model high-order interactions among brain regions of interest (ROIs) for neurodegenerative disease classification. The method learns continuous, scale-aware hyperedges rather than relying on predefined hyperedges, and integrates a scale-wise self-attention transformer to aggregate information across resolutions. Experiments on ADNI (5-class classification) and PPMI (3-class classification) show consistent improvements over graph-based and hypergraph-based baselines.

**Compliance With Llm Reviewing Policy:**

Affirmed.

**Final Justification:**

My concerns were properly tackled. I have kept my score as Accept because I believe it's an interesting work, but I have not worked with hypergraph architectures, nor I know enough about this field to comment on whether this work can be seen as "Accept" or "Strong Accept".

**Key Questions For Authors:**

Any counter argument to the weakness I’ve identified will be appreciated .

**Limitations:**

The authors identify a potential problem with this model in the Impact Statement (potential additional computation), but this is really no potential negative societal impact. Given this is a paper with potential clinical impact, I find this to not be correct. Easy starting points of discussion would be the acknowledgement that deployment on underrepresented states carries higher risk (see for example only 12 AD people), and a cautionary note on the interpretability claims that could be over-interpreted clinically if taken as causal rather than correlational biomarkers.

**Strengths And Weaknesses:**

# Strengths

I think the paper brings a well-motivated formulation, coupling graph wavelet decomposition with hyperedges construction, giving the model a concrete inductive bias to explore.

The experiments are quite thorough, with 5-fold cross validation on two independent and clinically-relevant datasets, with many baselines spanning both graph and hypergraph models, zero-shot transfer, dual-modality settings, stability analysis across folds, and others.

Finally, the interpretability analysis are good, and even though I cannot confirm whether the identified hub ROIs align with established literature, at least they are presented in a way that I believe would be enough for a clinician to analyse.

I must say that I think this is a very good work for ICML, but this paper would be much better places in a journal where more than half of the content wouldn’t be in appendix.


# Weaknesses
1. A key weakness that I find is that the hyperparameter search space is defined for MuHL, but not for all the other models, nor the number of parameters for each model is disclosed, to understand potential effects of model capacity. Given such imbalanced datasets, on both PPMI and AD (for example it seems there are only 12 people with AD?), one would imagine that the use of regularisations would be of extreme importance for a fair comparison. The fact that these results are so consistent, makes one wonder whether there is an influence from  a fair hyperparameter search.
2. In figure 5 it’s not clear whether we are looking at boxplots or, if just means, whether the sharded area corresponds to standard deviation or something else.
3. The prodromal cohort in PPMI can be seen as a mix of "healthy" and future PD people, as it basically includes anyone with a risk to develop PD. In this sense, the relevance of this 3-class classification is not as significant given the actual mix with the other cohorts, and might explain the not so good overall results here.

---

> ### Author Rebuttal · Authors · 2026-03-27
>
> We really appreciate the reviewer dcFP for the thoughtful and constructive reviews. We are especially thankful for the reviewer’s positive assessment, including strengths, presentation, significance, and originality, and we will clear out all the remaining questions.
>
> ---
>
> **[W1] Fairness of hyperparameter search**
>
> [A] We would like to clarify that the hyperparameter search was not limited to MuHL or designed to favor our model. As described in Appendix C, we report the key search settings for all compared methods and make comparable efforts to optimize both MuHL and the baselines. For fairness, the major factors that strongly affect generalization on small and imbalanced datasets, including hidden dimension, dropout rate, and learning rate, were controlled consistently across models within each dataset, while model-specific hyperparameters were tuned separately for each method. Since overfitting control is especially important in this regime, dropout, weight decay, and early stopping were handled carefully across all models. By providing an equivalent level of optimization effort to all methods, we intended for the observed performance gains to reflect fundamental architectural differences rather than an uneven search effort or tuning advantage.
>
> ---
>
> **[W2] Statistical visualization in Fig. 5**
>
> [A] We thank the reviewer for pointing this out. Fig. 5 does not show boxplots. The central box represents the mean value, and the shaded region indicates the standard deviation across folds. We agree that this was not sufficiently clear in the current caption and figure description. We will revise both to explicitly explain the meaning of the box and the shaded area.
>
> ---
>
> **[W3] Clinical significance of PPMI setting**
>
> [A] We agree that the prodromal cohort is inherently heterogeneous and can include subjects closer to either healthy controls or future PD. However, this ambiguity is precisely what makes the 3-class classification clinically meaningful and challenging. The goal of including the prodromal group is not to create an artificially clean classification problem, but to test whether the model can capture subtle transitional patterns that arise in realistic disease progression. Therefore, the lower overall performance on PPMI reflects the intrinsic biological difficulty of identifying pre-symptomatic states rather than a weakness in the experimental design. By including this challenging cohort, we aim to provide a more realistic assessment of how neuroimaging models perform in complex, real-world clinical scenarios.
>
> ---
>
> **[L1] Clinical risks and interpretational caution in the Impact Statement**
>
> [A] We thank the reviewer for this valuable suggestion. We agree that the current Impact Statement should better reflect the risks associated with clinical deployment. If the paper is accepted, we will revise it to state more explicitly that performance may be less reliable for underrepresented disease groups and that the model’s highlighted ROIs and hyperedges should not be overinterpreted as causal biomarkers. We will further clarify that these findings should be treated as correlational and decision-supportive evidence, and that clinical use would require careful human oversight.

---

> > ### Author Rebuttal · Reviewer_dcFp · 2026-04-02
> >
> > I thank the authors for their rebuttal. My concerns were properly tackled, and I apologise it seems I somehow missed appendix C.
> >
> > I'd just further say that I don't completely agree with the authors answer on W3. Having a challenging dataset is different from having a clinically relevant dataset. The fact that the prodromal could either be healthy, clinically diagnosed with PD or other diseases (in the future), makes a classification for the "prodromal label" not clinically relevant, and thus less significant.
> >
> > For now I have to keep to my score as Accept because I believe it's an interesting work, but I have not worked with hypergraph architectures, nor I know enough about this field to comment on whether this work can be seen as "Accept" or "Strong Accept". Because of this, I'll wait for the remaining reviewers before making a decision on whether to increase or not my score.

---

### Official Review · Reviewer_jzNC · 2026-03-09

**Soundness:** 3
**Presentation:** 3
**Significance:** 3
**Originality:** 3
**Overall Recommendation:** 4
**Confidence:** 3

**Summary:**

This paper introduces MuHL (Multi-scale Hyperedge Learning) , a novel framework for analyzing brain networks. Traditional Graph Neural Networks (GNNs) focus on pairwise connections between brain regions (nodes), failing to capture complex, higher-order interactions involving multiple regions simultaneously. MuHL addresses this by combining multi-resolution graph wavelet filtering with dynamic hypergraph learning to model these group-wise dependencies effectively.

**Compliance With Llm Reviewing Policy:**

Affirmed.

**Key Questions For Authors:**

See Weakness

**Limitations:**

Yes

**Strengths And Weaknesses:**

## strengths:
1.  The paper's primary strength is its novel integration of spectral graph wavelets with learnable hypergraph construction. This moves beyond simple pairwise models and predefined hypergraphs, offering a more flexible and powerful way to model complex brain networks.

2.  A key strength is its ability to provide clinically and biologically meaningful insights. By analyzing the learned hyperedges, the model successfully identified well-known disease-related brain regions, making its predictions more trustworthy and useful for hypothesis generation.

3.  The paper includes thorough ablation studies, hyperparameter sensitivity analyses, and tests on modality combinations. This robustly validates the contribution of each model component (multi-scale filtering, hypergraph learning, transformer) and its behavior under different data conditions.

4. The authors provide mathematical proofs (e.g., linking hyperedge construction to wavelet coefficients) to support their methodology, grounding the empirical results in a strong theoretical framework.

## Weaknesses
 The proposed framework was evaluated on only two datasets (ADNI and PPMI), which are specific to neurodegenerative diseases (Alzheimer's and Parkinson's). While a brief experiment on the ABIDE dataset for autism is mentioned in the supplementary material, the primary findings and claims of superiority are derived from these two sources. This narrow scope limits the generalizability of the results, and further validation on a wider range of datasets across diverse neurological and psychiatric conditions is necessary to confirm the model's broad applicability and robustness.

---

> ### Author Rebuttal · Authors · 2026-03-27
>
> We sincerely appreciate Reviewer jzNC for the constructive comments. We are especially grateful that the reviewer recognized the key strengths of our method and its relevance to brain network modeling. Now, we will answer the questions and strengthen the reviewer’s understanding of our contribution.
>
> ---
>
> **[W1] Limited generalizability**
>
> [A] We would like to clarify that our evaluation is **not limited to two narrow sources**, as it covers a diverse range of clinical and technical scenarios. In addition to the two main neurodegenerative cohorts ADNI and PPMI, we also evaluated MuHL on ABIDE (in Appendix D), which extends the analysis to a neurodevelopmental condition and therefore goes beyond the AD and PD setting. Moreover, Appendix D (Table 7 and Fig. 6) provides further evidence of robustness through additional ADNI experiments under diverse modality combinations, showing that MuHL generalizes well across heterogeneous input configurations.
>
> More importantly, Table 2 includes **zero-shot experiments under practically important distribution shifts.** Specifically, MuHL was trained on one acquisition phase or cohort and directly tested on independent target data without fine-tuning, including cross-phase transfer within ADNI and cross-cohort transfer from PPMI to TaoWu and Neurocon. These settings are substantially more challenging than standard in-domain evaluations and directly address the reviewer's concern regarding broad applicability and robustness across different acquisition phases, scanners, and independent sites. As a result, the current experiments already cover neurodegenerative, neurodevelopmental, cross-phase, and cross-dataset scenarios, and therefore provide **broader evidence of robustness and applicability** of MuHL.

---

> > ### Author Rebuttal · Reviewer_jzNC · 2026-04-03
> >
> > The proposed framework was evaluated on only two datasets of neurodegenerative diseases including Alzheimer's and Parkinson's. The evaluation is limited. I keep my score.

---

### Official Review · Reviewer_1u9C · 2026-03-12

**Soundness:** 2
**Presentation:** 3
**Significance:** 2
**Originality:** 2
**Overall Recommendation:** 3
**Confidence:** 5

**Summary:**

This paper focuses on capturing high-order dependencies across multiple brain regions. To this end, it proposes a multi-resolution hyper-edge learning framework called MuHL. The method first applies learnable graph-wavelet filtering to obtain scale-dependent node representations, then constructs adaptive soft hyperedges through a learnable projection and sparsification procedure, and finally performs hypergraph convolution plus scale-wise self-attention for disease classification. Experiments are conducted on ADNI and PPMI, with additional zero-shot transfer, modality-ablation, and component-ablation studies. The paper also claims interpretability by highlighting ROIs and salient hyperedges associated with disease progression.

**Compliance With Llm Reviewing Policy:**

Affirmed.

**Final Justification:**

I maintain my original recommendation. The rebuttal is helpful in clarifying the authors' perspective and in explaining the practical difficulty of collecting neurodegenerative disease data. The main issues remain only partially addressed. The paper still appears closer to a reasonable extension of existing hypergraph-based modeling, and the notion of "multi-scale" remains largely model-driven rather than clearly grounded at the brain level. In terms of evaluation, while class imbalance and limited sample size are common in this domain, not all brain-network datasets are constrained to the same extent, and the current validation based on only two small, highly imbalanced datasets is still not strong enough to fully support the paper's empirical claims. Overall, the rebuttal provides useful context, but I am inclined to maintain my original score.

**Key Questions For Authors:**

1. The paper defines “multi-scale” through graph-wavelet diffusion on a single atlas. What is the intended neurobiological meaning of each scale, and how should readers interpret the learned scales in functional terms?
2. Is there any neuroscientific evidence to support the importance or discriminative value of the learned hyperedges?

**Limitations:**

The limitations discussed in the paper are mainly at the engineering level, such as the dataset-dependent choice of the number of scales and the increased memory and computational cost. However, a more fundamental limitation is that the proposed multi-scale design is introduced mainly from the model side rather than from the data side. As a result, the learned multi-scale structure lacks clear interpretability, especially in terms of whether different scales correspond to meaningful brain-network granularity.

**Strengths And Weaknesses:**

Strengths
1. The organization of this paper is clear and easy to follow.
2. The method is clearly presented, and the experiments are comprehensive, covering several aspects, including zero-shot transfer experiments, modality ablations, and component ablations. The appendix also discusses efficiency and dynamic-structure ablations.

Weaknesses
1. Novelty is somewhat limited. Although the paper is motivated by higher-order interaction modeling, the core idea still relies on hypergraph modeling. More specifically, the main difference from existing work is that it constructs adaptive soft hyperedges from wavelet-filtered multi-scale features, instead of using predefined hyperedges or only learning hyperedge weights. While this is a reasonable extension of existing hypergraph methods, the overall novelty appears modest.
2. The theoretical contribution is not especially strong. Proposition 1 is largely a reformulation showing that the incidence construction can be written in wavelet space. Proposition 2 only states that there exists at least one hyperedge whose dominant assignment expands as the scale increases. These results provide some support for the model design, but they do not explain why the learned multi-scale hypergraph captures task-relevant or biologically meaningful higher-order structure. As written, the theory feels more supportive than genuinely explanatory.
3. The “multi-scale” idea appears to be more model-driven than brain-driven, which limits interpretability. In this paper, different scales are defined through spectral diffusion on a single atlas, rather than through a neurobiologically grounded multi-resolution representation. As a result, it is unclear what each learned scale actually corresponds to in terms of meaningful functional granularity, which makes the brain-specific interpretation less convincing.
4. The interpretability claim is not fully convincing. Although the paper provides post-hoc visualizations of salient ROIs and hyperedges, it does not provide sufficient evidence to show that these highlighted hyperedges are truly important or meaningful for the prediction task.
5. The related work is incomplete. Both the general high-order methods and the brain-network-specific methods discussed in the paper are almost entirely limited to hypergraph-based approaches. However, there are many other methods for modeling the same type of higher-order dependencies, which are largely overlooked in the related-work section.
6. The experimental validation is still limited. The method is evaluated on only two small and highly imbalanced medical datasets. For example, ADNI uses a 5-way classification setting with only 12 AD subjects, and PPMI contains only 15 CN subjects versus 113 PD subjects. Given this setup, the reported performance gains are not fully convincing.

---

> ### Author Rebuttal · Authors · 2026-03-27
>
> We truly thank the reviewer 1u9C for helpful comments. Below, we address remaining concerns and hope that the clarifications will positively influence the reviewer’s assessment toward acceptance.
>
> ---
>
> **[W1] Novelty of MuHL**
>
> [A] We appreciate the reviewer’s concerns on novelty, and we would like to highlight the novelty and contribution of MuHL once again along with other reviewers’ opinions. Our main contribution is not the mere application of hypergraphs, but the joint formulation where scale-wise soft node-to-hyperedge assignments are learned directly from wavelet filtered features. This core idea was validated on brain network data, which soundly extracted group-wise characteristics of brain connectivity for AD and PD. Fortunately, all other reviewers evaluated our approach positively, specifically noting the coupling of multi-scale spectral filtering and adaptive hypergraph structure learning as a novel and well-motivated design. Unlike prior brain network hypergraph works that rely on static, predefined hyperedges or only refine hyperedge weights, MuHL makes higher-order structures both data-adaptive and scale-dependent. Thus, the novelty lies in this unified learning of interactions across scales, which takes a significant step forward from standard hypergraph modeling.
>
> ---
>
> **[W2] Theoretical contribution**
>
> [A] The primary role of Propositions 1 and 2 is to establish the soundness and the scale-dependent behavior of our proposed hypergraph construction, rather than to provide an exhaustive neurobiological explanation. Proposition 1 formalizes incidence construction in wavelet space, and Proposition 2 characterizes how hyperedge assignments expand with scale. These results provide principled support for the model design, while task relevance and biological plausibility are supported mainly through empirical results and visualizations.
>
> ---
>
> **[W3/Q1] Interpretability for multi-scale design**
>
> [A] In MuHL, each scale corresponds to a different diffusion range on the same fixed brain atlas, rather than to a separate atlas or a one-to-one functional subsystem. This design is intended to capture the inherent hierarchical organization of the brain. Smaller scales preserve more localized ROI variation, while larger scales capture broader interaction structure, reflecting the brain’s local segregation and global integration within a single anatomical framework. In this sense, the learned scales are best interpreted as a coarse-to-fine hierarchy of functional interaction patterns on one atlas.
>
> ---
>
> **[W4/Q2] Interpretability evidence**
>
> [A] In brain network modeling, a key objective is to identify interaction patterns that undergo task-relevant alterations, and the highlighted ROIs and hyperedges are meaningful in this sense because they are discriminative higher-order features learned directly through the classification objective. Most importantly, the support does not come only from individual ROI saliency, but also from the fact that the most important hyperedges are composed of regions already known to be involved in neurodegenerative disorders. We provide further discussion of the biological relevance of these hyperedges in our response to the reviewer Uken W2/Q1, suggesting MuHL captures disease-related organization at the network level, rather than isolated node importance alone. We will revise the paper to clarify that these findings support task-relevant and neuroscientifically plausible higher-order patterns, while remaining correlational rather than causal.
>
> ---
>
> **[W5] Scope of related work**
>
> [A] The current related works were intentionally organized around methods most directly comparable to our framework. Since MuHL is a hypergraph-based model, we prioritized approaches with similar higher-order modeling assumptions so that the comparison remains focused and meaningful. We agree that the broader literature extends beyond hypergraphs, and we will expand the discussion accordingly, including [1,2]. We would also be grateful for any specific recommended references from the reviewer, and will be happy to incorporate them.
>
> [1] Ling, Qinrui, et al. "High-order graphical topology analysis of brain functional connectivity networks using fMRI." 2025.
>
> [2] Choi, Seongjin, et al. "Hypergraph Neural Sheaf Diffusion: A Symmetric Simplicial Set Framework for Higher-Order Learning." 2025.
>
> ---
>
> **[W6] Experimental validity.**
>
> [A] We note that class imbalance is not an incidental weakness of our benchmark, but a realistic and inherent characteristic of clinical neuroimaging data. A model that performs well under such an imbalance is more practically valuable than one validated only on artificially balanced cohorts. For this reason, we emphasized precision, recall, and F1-score, which are more informative than accuracy. MuHL achieved consistent gains under this demanding protocol, supporting its robustness and practical utility for real-world medical datasets, where class imbalance routinely exists.

---

> > ### Author Rebuttal · Reviewer_1u9C · 2026-04-03
> >
> > While the rebuttal clarifies the authors’ intended positioning, my main concerns remain only partially resolved. I still find it difficult to view the overall method as a substantial methodological advance rather than a reasonable extension of existing hypergraph-based modeling. In addition, the notion of “multi-scale” remains primarily model-driven: the learned scales are induced by spectral diffusion on a single atlas, without neurobiologically grounded priors, which limits the strength of the brain-specific interpretability claim. Finally, I remain unconvinced by the current experimental validation. The main evaluation relies on only two small and highly imbalanced datasets, with very small minority classes. Although precision, recall, and F1 are more informative than accuracy under class imbalance, in this extremely imbalanced small-sample setting these metrics may also be unstable and therefore do not fully resolve the reliability concern. In my view, these concerns are not easily addressed within a short rebuttal, as they would require more substantial revision to both the framing and the empirical support of the paper, including stronger justification for the brain-specific interpretability claim and validation on more suitable datasets.

---

> > > ### Author Response · Authors · 2026-04-04
> > >
> > > i) We totally understand it even if the reviewer evaluates that our method doesn't bring a substantial methodological advance but rather a reasonable extension.
> > >
> > > ii) In terms of dataset characteristics, brain networks data are inherently limited; it costs substantial resources to collect the images, i.e., MRI, Diffusion MRI, Amyloid PET and FDG PET, and efforts to process the them (e.g., multi-modal registration, tractography, etc.) to constitute a complete multi-modal and multi-sensing data. Moreover, collecting diseased samples is much more difficult than healthy controls as the diseased people pass away. These are fundamental characteristics of datasets for studying neurodegerenative diseases in elders which pose extremely challenging scenarios for ML problems. Of course we can simply utilize CN (N=226), SMC (N=131), and EMCI (N=217) groups in the preclinical stages of ADNI with less imbalance, however, it makes the problem less interesting in both ML and Neuroscientific perspectives.
> > >
> > > Moreover, as the reviewer already knows, the robustness of the experiments come from the "central limit theorem" that sum of random variables follow Gaussian distribution, and we performed experiments with cross-validation under various scenarios. In the end, notice that if the model was overfitted to the larger classes, then the precision for smaller classes, e.g., LMCI or AD, should have been close to 0 significantly damaging the precision, but it did not happen with our framework. Our approach achieved superior performance than the baselines "on average", which let us conclude that our model is empirically  demonstrating better performance with statistical evidence.
> > >
> > > Therefore, we strongly argue that the dataset cannot be a critical issue for the paper to be negatively valued, and if the setup is an issue, then there would be no valuable work for developing ML frameworks for studying neurodegenerative diseases.

---

### Decision · Program_Chairs · 2026-04-30

**Decision:**

Accept (regular)

**Comment:**

The authors present an end to end framework that jointly performs multi-resolution graph wavelet filtering and hypergraph learning to model higher-order relationships in brain networks. Reviewers appreciated formulation of the method, theoretical results supporting model design (even though they do not guarantee meaningful neurobiological explanation), clarity of the paper and the comprehensiveness of the experiments. Specifically the authors performed multiple experiments including  cross-phase, and cross-dataset scenarios, zero shot learning, reduced modality as well as ablation, sensitivity, and robustness studies. Biological interpretability of the studies was also highlighted by some of the reviewers.

Two reviewers expressed concerns regarding the size of the dataset, however the AC judges the experiments to be appropriate for connectivity analysis in neuroscience (the paper is within the neuroscience applications track), and the breadth of experiments using the three datasets is comprehensive. ADNI and PPMI are popular benchmarks in the field and the class imbalance is reflective of disease prevalence in the population.

For the final version of the paper the authors should carefully read the rebuttal and incorporate changes requested by reviewers. Specifically they need to clarify, among other issues, that these findings support task-relevant and neuroscientifically plausible higher-order patterns but these results are not causal, extend the related work to cover the field of higher-order analysis for brain network more broadly,  and the impact Statement needs to be updated to reflect the risks associated with clinical deployment.